# Generalised $f$-Mean Aggregation for Graph Neural Networks

**Ryan Kortvelesy, Steven Morad and Amanda Prorok**
University of Cambridge
{rk627, sm2558, asp45}@cam.ac.uk

## Abstract

Graph Neural Network (GNN) architectures are defined by their implementations of update and aggregation modules. While many works focus on new ways to parametrise the update modules, the aggregation modules receive comparatively little attention. Because it is difficult to parametrise aggregation functions, currently most methods select a "standard aggregator" such as mean, sum, or max. While this selection is often made without any reasoning, it has been shown that the choice in aggregator has a significant impact on performance, and the best choice in aggregator is problem-dependent. Since aggregation is a lossy operation, it is crucial to select the most appropriate aggregator in order to minimise information loss. In this paper, we present GenAgg, a generalised aggregation operator, which parametrises a function space that includes all standard aggregators. In our experiments, we show that GenAgg is able to represent the standard aggregators with much higher accuracy than baseline methods. We also show that using GenAgg as a drop-in replacement for an existing aggregator in a GNN often leads to a significant boost in performance across various tasks.

## 1 Introduction

Graph Neural Networks (GNNs) provide a powerful framework for operating over structured data. Taking advantage of relational inductive biases, they use local filters to learn functions that generalise over high-dimensional data. Given different graph structures, GNNs can represent many special cases, including CNNs (on grid graphs) [12], RNNs (on line graphs) [4], and Transformers (on fully connected graphs) [19]. All of these architectures can be subsumed under the Graph Networks framework, parametrised by update and aggregation modules [2]. Although the framework itself is general, the representational capacity is often constrained in practice through design choices, which create a human prior [21]. There are two primary reasons for introducing this human prior. First, there are no standard methods to parametrise all of the modules—MLPs can be used as universal approximators in the update modules, but it is nontrivial to parametrise the function space of aggregators. Consequently, most GNNs simply make a design choice for the aggregation functions, selecting mean, sum, or max [21].

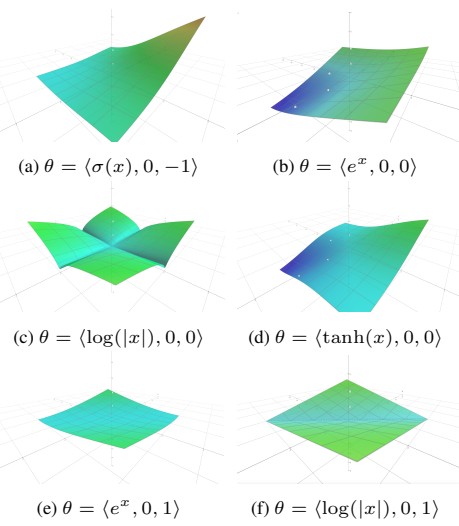

(a) $\theta = \langle \sigma(x), 0, -1 \rangle$    (b) $\theta = \langle e^x, 0, 0 \rangle$

(c) $\theta = \langle \log(|x|), 0, 0 \rangle$    (d) $\theta = \langle \tanh(x), 0, 0 \rangle$

(e) $\theta = \langle e^x, 0, 1 \rangle$    (f) $\theta = \langle \log(|x|), 0, 1 \rangle$

Figure 1: A qualitative demonstration of the diversity of functions that can be represented by GenAgg. In these visualisations, GenAgg is plotted as a function of inputs $x_0$ and $x_1$ for different parametrisations $\theta = \langle f, \alpha, \beta \rangle$ (see Equation (1)).

37th Conference on Neural Information Processing Systems (NeurIPS 2023).

Second, constraints can boost performance in GNNs, either through invariances or a regularisation effect.

In this paper, we focus on the problem of parametrising the space of aggregation functions. The ultimate goal is to create an aggregation function which can represent the set of all desired aggregators while remaining as constrained as possible. In prior work, one approach is to introduce learnable parameters into functions that could parametrise min, mean, and max, such as the Powermean and a variant of Softmax [8, 13, 20]. However, these approaches can only parametrise a small set of aggregators, and they can introduce instability in the training process (see Section 4). On the opposite end of the spectrum, methods like Deep Sets [22] and LSTMAgg [9] are capable of universal approximation over set functions, but they are extremely complex, which leads to poor sample efficiency. These methods scale in complexity (*i.e.* number of parameters) with the dimensionality of the input, and lack some of the useful constraints that are shared among standard aggregators (see Section 4). Consequently, the complexity of these methods counteracts the benefits of simple GNN architectures.

Although existing approaches present some limitations, the theoretical advantages of a learnable aggregation module are evident. It has been shown that the choice of aggregation function not only has a significant impact on performance, but also is problem-specific [21]. Since there is no aggregator that can discriminate between all inputs [5], it is important to select an aggregator that preserves the relevant information. In this paper, we present a method that parametrises the function space, allowing GNNs to learn the most appropriate aggregator for each application.

**Contributions**

- We introduce Generalised Aggregation (GenAgg), the first aggregation method based on the *generalised f-mean*. GenAgg is a learnable permutation-invariant aggregator which is provably capable (both theoretically and experimentally) of representing all "standard aggregators" (see Appendix C for proofs). The representations learned by GenAgg are *explainable*—each learnable parameter has an interpretable meaning (Section 6).

- Our experiments provide several insights about the role of aggregation functions in the performance of GNNs. In our regression experiments, we demonstrate that GNNs struggle to "make up for" the lack of representational complexity in their constituent aggregators, even when using state-of-the-art parametrised aggregators. This finding is validated in our GNN benchmark experiments, where we show that a GenAgg-based GNN outperforms all of the baselines, including standard aggregators and other state-of-the-art parametrised aggregators.

- Finally, we show that GenAgg satisfies a generalisation of the distributive property. We derive the solution for a binary operator that satisfies this property for given parametrisations of GenAgg. The generalised distributive property can be leveraged in algorithms using GenAgg to improve space and memory-efficiency.

## 2   Problem Statement

Consider a multiset $\mathcal{X} = \{x_1, x_2, \ldots, x_n\}$ of cardinality $|\mathcal{X}| = n$, where $x_i \in \mathbb{R}^d$. We define an aggregation function as a symmetric function $\bigodot : \mathbb{R}^{n \times d} \mapsto \mathbb{R}^{1 \times d}$. The aggregator must be independent over the feature dimension, so without loss of generality it can be represented over a single dimension $\bigodot : \mathbb{R}^n \mapsto \mathbb{R}^1$. A set of standard aggregators is defined $\mathcal{A} = \{\text{mean}, \text{sum}, \text{product}, \text{min}, \text{max}, \ldots\}$ (for the full list see Table 1). Our task is to create an aggregator $\bigoplus_\theta : \mathbb{R}^n \mapsto \mathbb{R}^1$ parametrised by $\theta$ which can represent all standard aggregators: $\forall \bigodot_i \in \mathcal{A} \; \exists \theta : \; \bigoplus_\theta = \bigodot_i$.

## 3   Method

In this section we introduce GenAgg, a parametrised aggregation function which is based on the *generalised f-mean* [11]. In our formulation, we introduce additional parameters to increase the representational capacity of the $f$-mean, producing the *augmented f-mean* (AFM). Then, as the implementation is non-trivial, we propose a method to implement it. The novel aspect of GenAgg is not

| Aggregation Function | $\alpha$ | $\beta$ | $f$ | GenAgg | SoftmaxAgg | PowerAgg |
|---|---|---|---|---|---|---|
| mean: $\frac{1}{n}\sum x_i$ | 0 | 0 | $f(x) = x$ | ✓ | ✓ | ✓ |
| sum: $\sum x_i$ | 1 | 0 | $f(x) = x$ | ✓ | ✗ | ✗ |
| product: $\prod \lvert x_i \rvert$ | 1 | 0 | $f(x) = \log(\lvert x \rvert)$ | ✓ | ✗ | ✗ |
| min (magnitude): $\min \lvert x_i \rvert$ | 0 | 0 | $f(x) = \lim_{p \to \infty} \lvert x \rvert^{-p}$ | ✓ | ✗ | ✓ |
| max (magnitude): $\max \lvert x_i \rvert$ | 0 | 0 | $f(x) = \lim_{p \to \infty} \lvert x \rvert^{p}$ | ✓ | ✗ | ✓ |
| min: $\min x_i$ | 0 | 0 | $f(x) = \lim_{p \to \infty} e^{-px}$ | ✓ | ✓ | ✗ |
| max: $\max x_i$ | 0 | 0 | $f(x) = \lim_{p \to \infty} e^{px}$ | ✓ | ✓ | ✗ |
| harmonic mean: $\frac{n}{\sum \frac{1}{x_i}}$ | 0 | 0 | $f(x) = \frac{1}{x}$ | ✓ | ✗ | ✓ |
| geometric mean: $\sqrt[n]{\prod \lvert x_i \rvert}$ | 0 | 0 | $f(x) = \log(\lvert x \rvert)$ | ✓ | ✗ | ✓ |
| root mean square: $\sqrt{\frac{1}{n}\sum x_i^2}$ | 0 | 0 | $f(x) = x^2$ | ✓ | ✗ | ✓ |
| euclidean norm: $\sqrt{\sum x_i^2}$ | 1 | 0 | $f(x) = x^2$ | ✓ | ✗ | ✗ |
| standard deviation: $\sqrt{\frac{1}{n}\sum (x_i - \mu)^2}$ | 0 | 1 | $f(x) = x^2$ | ✓ | ✗ | ✗ |
| log-sum-exp: $\log\left(\sum e^{x_i}\right)$ | 1 | 0 | $f(x) = e^x$ | ✓ | ✗ | ✗ |

Table 1: A table of all of the most common aggregators. For each special case, we specify the values of $\alpha$, $\beta$, and $f$ for which the augmented $f$-mean is equivalent (see Appendix C). We also report whether or not SoftmaxAgg and PowermeanAgg can represent each aggregator.

only the augmented $f$-mean formula, but also the implementation, which allows the mathematical concept of a generalised mean to be utilised in a machine learning context.

## 3.1 Generalised $f$-Mean

The *generalised f-mean* [11] is given by: $f^{-1}(\frac{1}{n}\sum_i f(x_i))$. While it is difficult to define aggregation functions, the generalised $f$-mean provides a powerful intuition: most aggregators can be represented by a single invertible scalar-valued function $f : \mathbb{R}^1 \mapsto \mathbb{R}^1$. This is a useful insight, because it allows comparisons to be drawn between aggregators by analysing their underlying functions $f$. Furthermore, it provides a framework for discovering new aggregation functions. While classic functions like $e^x, \log(x), x^p$ all map to aggregators in $\mathcal{A}$, new aggregators can be created by defining new functions $f$.

## 3.2 Augmented $f$-Mean

The standard generalised $f$-mean imposes strict constraints, such as symmetry (permutation-invariance), idempotency ($\bigoplus(\{x, \ldots, x\}) = x$), and monotonicity ($\forall i \in [1..n], \frac{\partial \bigoplus(\{x_1, \ldots, x_n\})}{\partial x_i} \geq 0$). However, this definition is too restrictive to parametrise many special cases of aggregation functions. For example, sum violates idempotency ($\sum_{i \in [1..n]} x_i = nx$), and standard deviation violates monotonicity ($\frac{\partial \sigma(\{x_1, 1\})}{\partial x_1}\big|_{x_1=0} < 0$). Consequently, we deem these constraints to be counterproductive. In our method, we introduce learnable parameters $\alpha$ and $\beta$ to impose a relaxation on the idetempotency and monotonicity constraints, while maintaining symmetry. We call this relaxed

formulation the *augmented f-mean* (AFM), given by:

$$\bigoplus_{i \in [1..n]} x_i = f^{-1}\left(n^{\alpha-1} \sum_{i \in [1..n]} f(x_i - \beta\mu)\right). \tag{1}$$

The $\alpha$ parameter allows AFM to control its level of dependence on the cardinality of the input $\mathcal{X}$. For example, given $f(x) = \log(|x|)$, if $\alpha = 0$, then AFM represents the geometric mean: $\bigoplus_{\langle f,\alpha,\beta\rangle} = \bigoplus_{\langle \log(|x|),0,0\rangle} = \sqrt[n]{\prod |x_i|}$. However, if $\alpha = 1$, then the $n$-th root disappears, and AFM represents a product: $\bigoplus_{\langle \log(|x|),1,0\rangle} = \prod |x_i|$.

The $\beta$ parameter enables AFM to calculate *centralised moments*, which are quantitative measures of the distribution of the input $\mathcal{X}$ [17]. The first raw moment of $\mathcal{X}$ is the mean $\mu = \frac{1}{n}\sum x_i$, and the $k$-th central moment is given by $\mu_k = \sum (x_i - \mu)^k$. With the addition of $\beta$, it becomes possible for AFM to represent $\sqrt[k]{\mu_k}$, the $k$-th root of the $k$-th central moment. For $k = 2$, this quantity is the standard deviation, which is in our set of standard aggregators $\mathcal{A}$. If the output is scaled to the $k$-th power, then it can also represent metrics such as variance, unnormalised skewness, and unnormalised kurtosis. It is clear that these metrics about the distribution of data are useful—they can have real-world meaning (*e.g.*, moments of inertia), and they have been used as aggregators in GNNs in prior work [5]. Consequently, $\beta$ provides AFM with an important extra dimension of representational complexity. In addition to representing the centralised moments when $\beta = 1$ and $f(x) = x^p$, $\beta$ allows *any* aggregator to be calculated in a centralised fashion. While the centralised moments are the only well-known aggregators that arise from nonzero $\beta$, there are several aggregators with qualitatively unique behaviour that can only be represented with nonzero $\beta$ (see Fig. 1).

With this parametrisation, AFM can represent any standard aggregator in $\mathcal{A}$ (Table 1). Furthermore, by selecting new parametrisations $\theta = \langle f, \alpha, \beta\rangle$, it is possible to compose new aggregators (Fig. 1).

## 3.3 Implementation

In Equation (1), the manner in which $f^{-1}$ is implemented is an important design choice. One option is to learn the coefficients for an analytical function (*e.g.* a truncated Taylor Series) and analytically invert it. However, it can be difficult to compute the analytical inverse of a function, and without carefully selected constraints, there is no guarantee that $f$ will be invertible.

Another possible option is an invertible neural network (*e.g.* a parametrised invertible mapping from *normalising flows* [10]). We have tested the invertible networks from normalising flows literature as implementations for $f$. While they work well on smaller tasks, these methods present speed and memory issues in larger datasets.

In practice, we find that the most effective approach is to use two separate MLPs for $f$ and $f^{-1}$. We enforce the constraint $x = f^{-1}(f(x))$ by minimizing the following optimisation objective:

$$\mathcal{L}_{\text{inv}}(\theta_1, \theta_2) = \mathbb{E}\left[\left(\left|f_{\theta_2}^{-1}(f_{\theta_1}(x))\right| - |x|\right)^2\right]. \tag{2}$$

The absolute value operations apply a relaxation to the constraint, allowing $f^{-1}(f(x))$ to reconstruct either $x$ or $|x|$. This is useful because several of the ground truth functions from Table 1 include an absolute value, making them non-invertible. With this relaxation, it becomes possible to represent those cases. This optimisation objective ensures that $f$ is both monotonic and invertible over the domains $\mathbb{R}^+$ and $\mathbb{R}^-$, independently. In our implementation, this extra optimisation objective is hidden behind the GenAgg interface and gets applied automatically with a forward hook, so it is not necessary for the user to apply an extra loss term.

While using a scalar-valued $f : \mathbb{R}^1 \mapsto \mathbb{R}^1$ is the most human-interpretable formulation, it is not necessary. A valid implementation of GenAgg can also be achieved with $f : \mathbb{R}^1 \mapsto \mathbb{R}^d$ and $f^{-1} : \mathbb{R}^d \mapsto \mathbb{R}^1$. In our experiments, we found that mapping to a higher intermediate dimension can sometimes improve performance over a scalar-valued $f$ (see training details in Appendix E).

## 3.4 Generalised Distributive Property

Given that GenAgg presents a method of parameterising the function space of aggregators, it can also be used as a tool for mathematical analysis. To demonstrate this, we use the aug-

| Aggregation Function | Distributive Operations $\psi(a,b)$ |
|---|---|
| mean: $\frac{1}{n}\sum x_i$ | $a+b, \quad a \cdot b$ |
| sum: $\sum x_i$ | $a \cdot b$ |
| product: $\prod \lvert x_i \rvert$ | $\lvert a \rvert^{\log \lvert b \rvert}$ |
| min (magnitude): $\min \lvert x_i \rvert$ | $\min(\lvert a \rvert, \lvert b \rvert)$ |
| max (magnitude): $\max \lvert x_i \rvert$ | $\max(\lvert a \rvert, \lvert b \rvert)$ |
| min: $\min x_i$ | $\min(a,b)$ |
| max: $\max x_i$ | $\max(a,b)$ |
| harmonic mean: $\dfrac{n}{\sum \frac{1}{x_i}}$ | $\frac{a \cdot b}{a+b}, \quad a \cdot b$ |
| geometric mean: $\sqrt[n]{\prod \lvert x_i \rvert}$ | $\lvert a \rvert \cdot \lvert b \rvert, \quad \lvert a \rvert^{\log \lvert b \rvert}$ |
| root mean square: $\sqrt{\frac{1}{n}\sum x_i^2}$ | $\sqrt{a^2 + b^2}, \quad \lvert a \rvert \cdot \lvert b \rvert$ |
| euclidean norm: $\sqrt{\sum x_i^2}$ | $\lvert a \rvert \cdot \lvert b \rvert$ |
| standard deviation: $\sqrt{\frac{1}{n}\sum (x_i - \mu)^2}$ | $\lvert a \rvert \cdot \lvert b \rvert$ |
| log-sum-exp: $\log \left( \sum e^{x_i} \right)$ | $a + b$ |

Table 2: A table of the distributive operations $\psi$ that satisfy each aggregation function, computed using Equation 3. All aggregation functions have at least one solution, and some special cases have multiple solutions.

mented $f$-mean to analyse a generalised form of the distributive property, which is satisfied if $\psi\left(c, \bigodot_{x_i \in \mathcal{X}} x_i\right) = \bigodot_{x_i \in \mathcal{X}} \psi(c, x_i)$ for binary operator $\psi$ and aggregator $\bigodot$. For a given aggregation function parametrised by $f$ (assuming $\beta$ is 0), we derive a closed-form solution for a corresponding binary operator which will satisfy the generalised distributive property (for further explanation and proofs, see Appendix A).

**Theorem 3.1.** *For GenAgg parametrised by $\theta = \langle f, \alpha, \beta \rangle = \langle f, \alpha, 0 \rangle$, the binary operator $\psi$ which will satisfy the Generalised Distributive Property for $\bigoplus_\theta$ is given by:*

$$\psi(a,b) = f^{-1}(f(a) \cdot f(b)) \tag{3}$$

*Furthermore, for the special case $\theta = \langle f, \alpha, \beta \rangle = \langle f, 0, 0 \rangle$, there $\psi(a,b) = f^{-1}(f(a) + f(b))$ is an additional solution.*

For example, for the euclidean norm where $f(x) = x^2$ and $\alpha = 1$, the binary operator is $\psi(a,b) = (a^2 \cdot b^2)^{\frac{1}{2}} = a \cdot b$, which implies that a constant multiplicative term can be moved outside of the euclidean norm. This is a useful finding, as the distributive property can used to improve algorithmic time and space complexity (*e.g.* the FFT) [1]. With our derivation of $\psi$ as a function of $f$, it is possible to implement similar efficient algorithms with GenAgg.

## 4 Related Work

Several existing works propose methods to parametrise the space of aggregation functions. These methods can broadly be divided into two categories. *Mathematical* approaches derive an explicit equation in terms of the inputs and one or more learnable parameters. Usually, these approaches represent a smooth interpolation through function space from min, through mean, to max. Alternatively, *Deep Learning* approaches seek to use the universal approximation properties of neural networks to maximise representational complexity.

### 4.1 Mathematical Approaches

**SoftmaxAgg** SoftmaxAgg computes the weighted sum of the set, where the weighting is derived from the softmax over the elements with some learnable temperature term $s$ [13, 20]. This formu-

lation allows SoftmaxAgg to represent mean, min, and max (see Table 1). Unfortunately, it fails to generalise across the majority of the standard aggregators.

**PowerAgg** Based on the $p$-norm, PowerAgg is a special case of GenAgg where $\alpha = 0$, $\beta = 0$, and $f(x) = x^p$. There are some methods which use the powermean directly [13, 20, 8], and others which build on top of it [18]. Theoretically, PowerAgg can represent a significant subset of the standard aggregators: min magnitude, max magnitude, mean, root mean square, harmonic mean, and geometric mean (although the geometric mean requires $\lim_{p \to 0}$, so it is not practically realisable) (see Table 1). Unfortunately, there is a caveat to this approach: for negative inputs $x_i < 0$ and non-integer values $p$, it is only defined in the complex domain. Furthermore, for negative inputs, the gradient $\frac{\partial x^p}{\partial p}$ with respect to trainable parameter $p$ is complex and oscillatory (and therefore is prone to getting stuck in local optima). In order to fix this problem, the inputs must be constrained to be positive. In prior work, this has been achieved by clamping $x_i' = \max(x_i, 0)$ [13], subtracting the minimum element $x_i' = x_i - \min(\mathcal{X})$ [20], or taking the absolute value $x_i' = |x_i|$ [8]. However, this removes important information, making it impossible to reconstruct most standard aggregators.

### 4.2 Deep Learning Approaches

**PNA** While Principle Neighbourhood Aggregation [5] is introduced as a GNN architecture, its novelty stems from its method of aggregation. In PNA, input signal is processed by a set of aggregation functions, which is produced by the cartesian product of standard aggregators {mean, min, max, std} and scaling factors $\{\frac{1}{n}, 1, n\}$. The output of every aggregator is concatenated, and passed through a dense network. While this increases the representational complexity of the aggregator, it also scales the dimensionality of the input by the number of aggregators multiplied by the number of scaling factors, which can decrease sample efficiency (Figure 3). Furthermore, the representational complexity of the method is limited by the choice of standard aggregators—it cannot be used to represent many of the special cases of parametrised general aggregators.

**LSTMAgg** In LSTMAgg, the input set is treated as a sequence (applying some random permutation), and is encoded with a recurrent neural network [9]. While this method is theoretically capable of universal approximation, in practice its non-permutation-invariance can cause its performance to suffer (as the factorial complexity of possible orderings leads to sample-inefficiency). SortAgg addresses this issue by sorting the inputs with computed features, and passing the first $k$ sorted inputs through convolutional and dense networks [23]. While this method solves the issue of non-permutation-invariance, it loses the capability of universal approximation by truncating to $k$ inputs. While universal approximation is not a requirement for an effective aggregation function, we note that it cannot represent many of the standard aggregators.

**Deep Sets** Deep Sets is a universal set function approximator [22]. However, because it operates over the feature dimension in addition to the "set" dimension, it is not regarded as an aggregation function. Instead, it usually serves as a full GNN layer or graph pooling architecture [14, 15]. One may note that the formulation for Deep Sets $\phi(\sum_{i \in [1..n]} f(x_i))$ bears some resemblance to our method. However, there are two important differences. First, our method adds the constraint $\phi = f^{-1}$, limiting possible parametrisations to the subspace where all of the standard aggregators lie. Second, while the learnable functions $\phi$ and $f$ in Deep Sets are fully connected over the feature dimension, the $f$ and $f^{-1}$ modules in our architecture are scalar-valued functions which are applied element-wise. To summarise, Deep Sets is useful as a set function approximator, but it lacks constraints that would make it viable as an aggregation function.

## 5 Experiments

In this paper, we run three experiments. First, we show that GenAgg can perform regression to recover any standard aggregation function. Then, we evaluate GenAgg and several baselines inside of a GNN. The resulting GNN architectures are given the same task of regressing upon graph-structured data generated with a standard aggregator. This tests if it is possible for a GNN with a given aggregator to represent data which was generated by different underlying aggregators. Finally, we provide practical results by running experiments on public GNN benchmark datasets: CLUSTER, PATTERN, CIFAR10, and MNIST [6].

| Aggregation | GenAgg | P-Agg | S-Agg | mean |
|---|---|---|---|---|
| mean | **1.000** | 0.817 | **1.000** | **1.000** |
| sum | **1.000** | 0.761 | 0.887 | 0.888 |
| product (mag) | **0.985** | 0.407 | 0.172 | 0.022 |
| min (mag) | **0.962** | 0.450 | 0.024 | 0.027 |
| max (mag) | **0.990** | 0.586 | 0.423 | 0.024 |
| min | **0.995** | 0.734 | **1.000** | 0.805 |
| max | **0.990** | 0.920 | **1.000** | 0.805 |
| harm. mean (abs) | **0.986** | 0.453 | 0.088 | 0.027 |
| geom. mean (abs) | **0.994** | 0.481 | 0.152 | 0.031 |
| root mean square | **0.996** | 0.532 | 0.308 | 0.028 |
| euclidean norm | **0.964** | 0.585 | 0.464 | 0.019 |
| standard dev. | **0.999** | 0.442 | 0.558 | 0.013 |
| log-sum-exp | **0.999** | 0.823 | 0.947 | 0.747 |

(a) Aggregator Regression.

| Aggregation | GenAgg | P-Agg | S-Agg | mean |
|---|---|---|---|---|
| mean | 0.977 | **0.999** | **1.000** | 0.972 |
| sum | **0.971** | 0.906 | 0.887 | 0.903 |
| product (mag) | **0.966** | 0.644 | 0.726 | 0.434 |
| min (mag) | **0.952** | 0.876 | 0.810 | 0.731 |
| max (mag) | **0.986** | 0.734 | 0.784 | 0.747 |
| min | **0.995** | 0.986 | **0.999** | 0.806 |
| max | **0.989** | 0.976 | **0.999** | 0.845 |
| harm. mean (abs) | **0.931** | 0.797 | 0.842 | 0.697 |
| geom. mean (abs) | **0.963** | 0.629 | 0.836 | 0.626 |
| root mean square | **0.975** | 0.775 | 0.808 | 0.899 |
| euclidean norm | **0.985** | 0.742 | 0.680 | 0.756 |
| standard dev. | **0.966** | 0.739 | 0.805 | 0.624 |
| log-sum-exp | **0.983** | 0.919 | 0.952 | 0.841 |

(b) GNN Regression

Figure 2: Results for the Aggregator Regression and GNN Regression experiments, indicating the ability of GenAgg, PowerAgg (P-Agg), SoftmaxAgg (S-Agg), and mean to parametrise each standard aggregator in $\mathcal{A}$. We report the correlation coefficient between the ground truth and predicted outputs. The highest-performing methods (and those within $0.01$ correlation) are shown in bold.

For all baselines, we use the implementations provided in PyTorch Geometric [7]. The only exception is PNA, which is a GNN architecture by nature, not an aggregation method. For our experiments, we adapt PNA into an aggregation method, staying as true to the original formulation as possible: $\text{PNA}(\mathcal{X}) = f([1, n, \frac{1}{n}] \otimes [\text{mean}(\mathcal{X}), \text{std}(\mathcal{X}), \min(\mathcal{X}), \max(\mathcal{X})])$, where $f$ is a linear layer mapping from $\mathbb{R}^{12d}$ back to $\mathbb{R}^d$.

For more training details, see Appendix E. Our code can be found at: `https://github.com/Acciorocketships/generalised-aggregation`.

## 5.1 Aggregator Regression

In this experiment, we generate a random graph $\mathcal{G} = \langle \mathcal{V}, \mathcal{E} \rangle$ with $|\mathcal{V}| = 8$ nodes and an edge density of $\frac{|\mathcal{E}|}{|\mathcal{V}|^2} = 0.3$. For each node $i \in \mathcal{V}$, we draw an internal state $x_i \in \mathbb{R}^d$ from a normal distribution $x_i \sim \mathcal{N}(\mathbf{0}_d, I_d)$ with $d = 6$. Then, we generate training data with a set of ground truth aggregators $\bigodot_k \in \mathcal{A}$ (where $k$ is an index). For each aggregator $\bigodot_k$, the dataset $X_k, Y_k$ is produced with a Graph Network [2], using $\bigodot_k$ as the node aggregation module. The inputs are defined by the set of neighbourhoods in the graph $X_k = \{\mathcal{X}_i \mid i \in [1..|\mathcal{V}|]\}$ where the neighbourhood $\mathcal{X}_i$ is defined as $\mathcal{X}_i = \{x_j \mid j \in \mathcal{N}_i\}$ with $\mathcal{N}_i = \{j \mid (i,j) \in \mathcal{E}\}$. The corresponding ground truth outputs are defined as $Y_k = \{y_i \mid i \in [1..|\mathcal{V}|]\}$, where $y_i = \bigodot_k(\mathcal{X}_i)$.

The model that we use for regression takes the same form as the model used to generate the data, except that the standard aggregator used to generate the training data $\bigodot_k$ is replaced with a parametrised aggregator $\bigoplus_\theta$:

$$\hat{y}_i = \bigoplus_{\substack{\theta \\ x_j \in \mathcal{X}_i}} x_j \tag{4}$$

In our experiments, each type of parametrised aggregator (GenAgg, SoftmaxAgg, PowerAgg, and mean as a baseline) is trained separately on each dataset $X_k, Y_k$.

**Results.** We report the MSE loss and correlation coefficient with respect to the ground truth in Table 2a. GenAgg is able to represent all of the standard aggregators with a correlation of at least $0.96$, and most aggregators with a correlation of greater than $0.99$. The only cases where the performance of GenAgg is surpassed by a baseline are min and max, where SoftmaxAgg exhibits marginally higher accuracy.

One interesting observation is that even if the baselines can represent an aggregator in *theory*, they cannot necessarily do so in practice. For example, PowerAgg can theoretically represent the geomet-

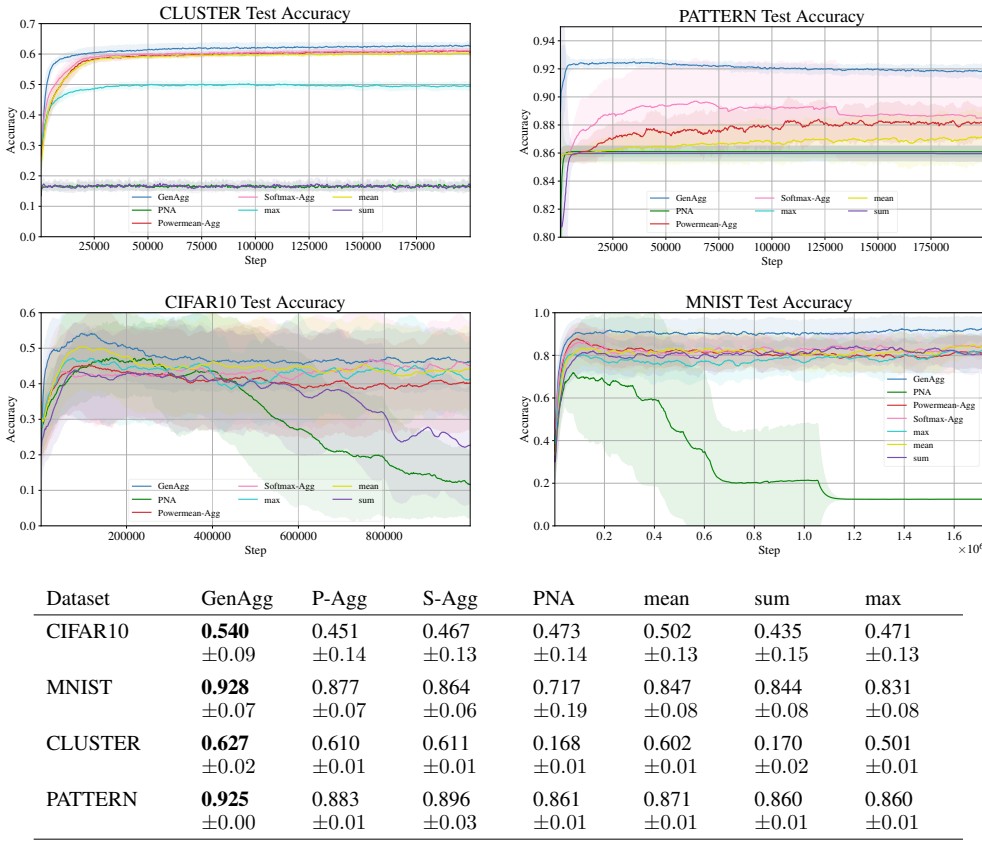

| Dataset | GenAgg | P-Agg | S-Agg | PNA | mean | sum | max |
|---------|--------|-------|-------|-----|------|-----|-----|
| CIFAR10 | **0.540** ±0.09 | 0.451 ±0.14 | 0.467 ±0.13 | 0.473 ±0.14 | 0.502 ±0.13 | 0.435 ±0.15 | 0.471 ±0.13 |
| MNIST | **0.928** ±0.07 | 0.877 ±0.07 | 0.864 ±0.06 | 0.717 ±0.19 | 0.847 ±0.08 | 0.844 ±0.08 | 0.831 ±0.08 |
| CLUSTER | **0.627** ±0.02 | 0.610 ±0.01 | 0.611 ±0.01 | 0.168 ±0.01 | 0.602 ±0.01 | 0.170 ±0.02 | 0.501 ±0.01 |
| PATTERN | **0.925** ±0.00 | 0.883 ±0.01 | 0.896 ±0.03 | 0.861 ±0.01 | 0.871 ±0.01 | 0.860 ±0.01 | 0.860 ±0.01 |

Figure 3: Test accuracy for GNNs with various aggregators on GNN benchmark datasets. In this experiment, each trial uses the same base GNN architecture (4-layer GraphConv), and the default aggregator is replaced with either GenAgg, PowermeanAgg (P-Agg), SoftmaxAgg (S-Agg), PNA, mean, sum, or max. The plots depict the mean and standard deviation of the test accuracy over 10 trials (note that the y-axis is scaled to increase readability). The table reports the maximum of the mean test accuracy over all timesteps, as well as the standard deviation (rounded to $0.01$).

ric mean with $\lim_{p\to 0}\left(\frac{1}{n}\sum_i x_i^p\right)^{\frac{1}{p}}$, but in practice there are instabilities as $p$ approaches 0 because $\frac{1}{p}$ approaches $\frac{1}{0}$. Similarly, while in theory PowerAgg can represent min magnitude, max magnitude, harmonic mean, and root mean square, it falls short in practice (see Table 2a), likely because of the reasons stated in Section 4. In other cases, the baselines can perform well even if they should not be able to represent the target in theory. One such example is PowerAgg, which achieves a correlation of $0.92$ on max, but only $0.59$ on max magnitude, which is the opposite of what theory might suggest. This is likely due to the the clamp operation that Pytorch Geometric's implementation uses to restricts inputs to the positive domain. The performance of max magnitude suffers, as it misses cases where the highest magnitude element is negative. Similarly, the performance of max increases, because it simply selects the maximum among the positive elements. Another baseline which performs unexpectedly well is SoftmaxAgg, which achieves a high correlation with the log-sum-exp aggregator. While it cannot compute a log, the SoftmaxAgg formulation does include a sum of exponentials, so it is able to produce a close approximation.

## 5.2 GNN Regression

In GNN Regression, the experimental setup is the same as that of Aggregator Regression (Section 5.1), with the exception that the observation size is reduced to $d = 1$. However, instead of using GenAgg $\bigoplus_\theta$ as a model, we use a multi-layer GNN. The GNN is implemented with 4 layers of GraphConv [16] with Mish activation (after every layer except the last), where the default aggrega-

tion function is replaced by a parametrised aggregator $\bigoplus_\theta$:

$$z_i^{(k+1)} = \text{Mish}\left(W_1^{(k)} z_i^{(k)} + W_2^{(k)} \bigoplus_{j \in \mathcal{N}_i}{}_\theta z_i^{(k)}\right), \text{ s.t. } z_i^{(0)} = x_i \tag{5}$$

$$\hat{y}_i = W_1^{(3)} z_i^{(3)} + W_2^{(3)} \bigoplus_{j \in \mathcal{N}_i}{}_\theta z_i^{(3)} \tag{6}$$

While this experiment uses the same dataset as Aggregator Regression (Section 5.1), it provides several new insights. First, while the aggregator regression experiment shows that GenAgg *can* represent various aggregators, this experiment demonstrates that training remains stable even when used within a larger architecture. Second, this experiment underlines the importance of using the correct aggregation function. While it is clear that it is advantageous to match a model's aggregation function with that of the underlying mechanism which generated a particular dataset, we often opt to simply use a default aggregator. The conventional wisdom of this choice is that the other learnable parameters in a network layer can rectify an inaccurate choice in aggregator. However, the results from this experiment demonstrate that even with additional parameters, it is not necessarily possible to represent a different aggregator, underlining the importance of aggregators with sufficient representational capacity.

**Results.** The results show that GenAgg maintains its performance, even when used as a component within a GNN (Table 2b). GenAgg achieves a mean correlation of $0.97$ across all aggregators. While the baselines perform significantly better with the help of a multi-layer GNN architecture, they still cannot represent many of the standard aggregators. The highest-performing baseline is SoftmaxAgg, which only achieves a mean correlation of $0.86$.

### 5.3 GNN Benchmark

In this experiment, we examine the performance of GenAgg on GNN benchmark datasets [6]. In order to perform a comparison with benchmarks, we train on an existing GNN architecture (a 4-layer GraphConv [16] GNN with a hidden size of 64) where the default aggregator is replaced with a new aggregator, selected from $\{\text{GenAgg}, \text{PowerAgg}, \text{SoftmaxAgg}, \text{PNA}, \text{mean}, \text{sum}, \text{max}\}$.

**Results.** As shown in Fig 3, GenAgg outperforms all baselines in all four GNN benchmark datasets. It provides a significant boost in performance, particularly compared to the relatively small differences in performance between the baseline methods.

The training plots in Fig. 3 provide complementary information. One interesting observation is that GenAgg converges at least as fast as the other methods, and sometimes converges significantly faster (in PATTERN, for example). Furthermore, the training plots lend information about the stability of training. For example, note that in MNIST, most of the baseline methods achieve a maximum and then degrade in performance, while GenAgg maintains a stable performance throughout training.

## 6 Discussion

**Results**. In our experiments, we present two regression tasks and one GNN benchmark task. The regression experiments demonstrate that GenAgg is the only method capable of representing all of the standard aggregators, and a GNN cannot be used to compensate for the shortcomings of the baseline aggregators. The GNN benchmark experiment complements these findings, demonstrating that this representational complexity is actually useful in practice. The fact that GenAgg outperforms the standard aggregators (mean, max, and sum) on the GNN benchmark experiment implies that it is in fact creating a *new* aggregator. Furthermore, the fact that it outperforms baseline methods like SoftmaxAgg and PowermeanAgg implies that the aggregator learned by GenAgg lies outside the set of functions which can be represented by such methods.

**Limitations**. While GenAgg achieves positive results on these datasets, it is not possible to make generalisations about its performance in all applications. In particular, we observe that some datasets fundamentally require less complexity to solve, so simple aggregators are sufficient (*i.e.*, GenAgg fails to provide a significant performance boost). For a full list of datasets that we considered and further discussion of limitations, see Appendix D.

**Parameters**. When comparing the performance of different models, it is important to also consider the number of parameters. By introducing additional parameters, some models can improve overall performance at the cost of sample efficiency. While methods like PowerAgg and SoftmaxAgg only have one trainable parameter, GenAgg has two scalar parameters $\alpha$ and $\beta$, and a learnable function $f$, which has 30 parameters in our implementation (independent of the size of the state). However, we observe that using GenAgg within a GNN is always at least as sample-efficient as the baselines, and sometimes converges significantly faster (Fig. 3 and Appendix B). Furthermore, while GenAgg has more parameters than PowerAgg and SoftmaxAgg, the increase is negligible compared to the total number of parameters in the GNN. We also note that GenAgg has significantly fewer parameters than the deep learning methods discussed in Section 4. While the deep learning methods scale linearly or quadratically in the dimension of the state, the number of parameters in GenAgg is constant.

**Stability**. Another observation from our experiments is that GenAgg exhibits more stability during the training process than the baselines (Appendix B). In the GNN Regression experiment, the PowerAgg and SoftmaxAgg training curves tend to plateau at least once before reaching their maximum value. It is possible that these methods lead to local optima because they are optimised in a lower dimensional parameter space [3]. For example, it is straightforward to smoothly transform a learned $f$ in GenAgg from $x^2$ to $x^4$, but to do so in PowerAgg, it is necessary to pass through $x^3$, which has significantly different behaviour in the negative domain. While PowerAgg restricts inputs to the positive domain to circumvent this particular issue, the problem of local optima can still arise when methods like PowerAgg or SoftmaxAgg are used as components in a larger architecture.

**Explainability**. While in this paper we primarily focus on the *performance* of GenAgg, we note that it also presents benefits in the realm of explainability. The three parameters in GenAgg are all human-readable (scalars and scalar-valued functions can easily be visualised), and they all provide a unique intuition. The $\alpha$ parameter controls the dependence on the cardinality of the input set. The $\beta$ parameter dictates if the aggregator is computed in a raw or centralised fashion (colloquially, it answers if the aggregator operates over the inputs themselves, or the variation between the inputs). Lastly, the function $f$ can be analysed by considering the sign and magnitude of $f(x_i)$. The sign denotes if a given $x_i$ increases ($f(x) > 0$) or decreases ($f(x_i) < 0$) the output. On the other hand, the magnitude $|f(x_i)|$ can be interpreted as the relative impact of that point on the output. For example, the parametrisation of product is $f(x) = \log(|x|)$, which implies that a value of 1 has no impact on the output since $|\log(|1|)| = 0$, and extremely small values $\epsilon$ have a large impact, because $\lim_{\epsilon \to 0} |\log(|\epsilon|)| = \infty$. Indeed, 1 is the identity element under multiplication, and multiplying by a small value $\epsilon$ can change the output by many orders of magnitude. The interpretability of GenAgg can also be leveraged as a method to *select* an aggregator—a model can be pre-trained with GenAgg, and then each instance of GenAgg can be replaced with the most similar standard aggregator in $\mathcal{A}$.

## 7 Conclusion

In this paper we introduced GenAgg, a generalised, explainable aggregation function which parametrises the function space of aggregators, yet remains as constrained as possible to improve sample efficiency and prevent overfitting. In our experiments, we showed that GenAgg can represent all 13 of our selected "standard aggregators" with a correlation coefficient of at least 0.96. We also evaluated GenAgg alongside baseline methods within a GNN, illustrating how other approaches have difficulties representing standard aggregators, even with the help of additional learnable parameters. Finally, we demonstrated the usefulness of GenAgg on GNN benchmark tasks, comparing the performance of the same GNN with various different aggregators. The results showed that GenAgg provided a significant boost in performance over the baselines in all four datasets. Furthermore, GenAgg often exhibited more stability and faster convergence than the baselines in the training process. These results show that GenAgg is an application-agnostic aggregation method that can provide a boost in performance as a drop-in replacement for existing aggregators.

## 8 Acknowledgements

Ryan Kortvelesy and Amanda Prorok were supported in part by ARL DCIST CRA W911NF-17-2-0181 and European Research Council (ERC) Project 949940 (gAIa).

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
