# Appendix

## A  Generalised Distributive Property

The Distributive Law is an extremely useful tool for analysis and computation in mathematics and computer science. Following from the additivity and homogeneity properties of linearity, it states that $\sum_{i=1}^{n} c \cdot x_i = c \cdot \sum_{i=1}^{n} x_i$. The Distributive Law is often leveraged to formulate fast algorithms, such as the Fast Fourier Transform and the Viterbi algorithm [1]. It can also be used to make algorithms more memory-efficient. For example, if we wish to take the DFT of a shifted signal, we can avoid storing the signal itself. Instead, the Distributive Law can be utilised to formulate the shifted DFT as a function of the non-shifted DFT: $\mathrm{DFT}[x(n-\Delta)] = e^{-i\omega_k \Delta} X(\omega_k)$.

While the Distributive Law is defined for the group of real numbers under additivity $(\mathbb{R}, +)$, we can also define a more general distributive property for the Abelian group defined by an aggregator $(\mathbb{R}, \odot)$.

**Definition A.1** (Generalised Distributive Property). For a binary operator $\psi$ and set aggregation function $\odot$, the Generalised Distributive Property is defined:

$$\psi \left( c, \bigodot_{x_i \in \mathcal{X}} x_i \right) = \bigodot_{x_i \in \mathcal{X}} \psi(c, x_i) \tag{7}$$

In this section, we derive the Generalised Distributive Property for the special case of GenAgg $\odot = \bigoplus_{\theta}$. That is, for a given parametrisation $\theta$, we derive an explicit formula for the corresponding function $\psi$ which satisfies the Generalised Distributive Property. Note that while our solution holds for any function $f$, it focuses on the special cases $\alpha = 0, \beta = 0$ and $\alpha = 1, \beta = 0$.

**Lemma A.1.** *Given a binary operator of the form $\psi(a, b) = \phi^{-1}(\phi(a) + \phi(b))$ and a parametrisation of GenAgg $\theta = \langle f, \alpha, \beta \rangle = \langle f, \alpha, 0 \rangle$, the operator $\psi(a, b)$ satisfies the Generalised Distributive Property over the $(\mathbb{R}, \oplus_{\theta})$ abelian group if:*

$$\rho^{-1} \left( \frac{n^{\alpha}}{n} \sum \rho(c + x_i) \right) = \rho^{-1} \left( \frac{n^{\alpha}}{n} \sum \rho(x_i) \right) + c \tag{8}$$

$$\text{where } \rho(x) = f(\phi^{-1}(x)) \tag{9}$$

*Proof.* Substituting GenAgg for the generic aggregation function $\odot$ in the definition of the Deneralised Distributive Property, we get:

$$f^{-1} \left( \frac{n^{\alpha}}{n} \sum f(\psi(c, x_i)) \right) = \psi \left( c, f^{-1} \left( \frac{n^{\alpha}}{n} \sum f(x_i) \right) \right) \tag{10}$$

We replace the binary operator $\psi$ with its representation as a composition of univariate functions $\psi(a, b) = \phi^{-1}(\phi(a) + \phi(b))$ to obtain:

$$f^{-1} \left( \frac{n^{\alpha}}{n} \sum f \left( \phi^{-1}(\phi(c) + \phi(x_i)) \right) \right) = \phi^{-1} \left( \phi \left( f^{-1} \left( \frac{n^{\alpha}}{n} \sum f(x_i) \right) \right) + \phi(c) \right) \tag{11}$$

$$\phi \left( f^{-1} \left( \frac{n^{\alpha}}{n} \sum f \left( \phi^{-1}(\phi(c) + \phi(x_i)) \right) \right) \right) = \phi \left( f^{-1} \left( \frac{n^{\alpha}}{n} \sum f(x_i) \right) \right) + \phi(c) \tag{12}$$

To simplify, we apply a change of variables $x' = \phi(x)$, $c' = \phi(c)$:

$$\phi \left( f^{-1} \left( \frac{n^{\alpha}}{n} \sum f \left( \phi^{-1}(c' + x_i') \right) \right) \right) = \phi \left( f^{-1} \left( \frac{n^{\alpha}}{n} \sum f(\phi^{-1}(x_i')) \right) \right) + c' \tag{13}$$

Finally, we further simplify by substituting $\rho(x) = f(\phi^{-1}(x))$:

$$\rho^{-1} \left( \frac{n^{\alpha}}{n} \sum \rho(c' + x_i') \right) = \rho^{-1} \left( \frac{n^{\alpha}}{n} \sum \rho(x_i') \right) + c' \tag{14}$$

$\square$

**Theorem A.2.** *For GenAgg parametrised by $\theta = \langle f, \alpha, \beta \rangle = \langle f, \alpha, 0 \rangle$, the binary operator $\psi$ which will satisfy the Generalised Distributive Property for $\bigoplus_\theta$ is given by $\psi(a, b) = f^{-1}(f(a) \cdot f(b))$.*

*Proof.* From Lemma A.1, the Generalised Distributive Property is satisfied if:

$$\rho^{-1}\left(\frac{n^\alpha}{n} \sum \rho(c + x_i)\right) = \rho^{-1}\left(\frac{n^\alpha}{n} \sum \rho(x_i)\right) + c \tag{15}$$

$$\text{where } \rho(x) = f(\phi^{-1}(x)) \tag{16}$$

If we select $\rho(x) = e^x$, then we can show that the condition is satisfied by the standard Distributive Law:

$$\log\left(\frac{n^\alpha}{n} \sum e^{c+x_i}\right) = \log\left(\frac{n^\alpha}{n} \sum e^{x_i}\right) + c \tag{17}$$

$$e^{\log\left(\frac{n^\alpha}{n} \sum e^{c+x_i}\right)} = e^{\log\left(\frac{n^\alpha}{n} \sum e^{x_i}\right) + c} \tag{18}$$

$$\frac{n^\alpha}{n} \sum e^c e^{x_i} = e^c \cdot \frac{n^\alpha}{n} \sum e^{x_i} \tag{19}$$

Given that $\rho(x) = e^x$ satisfies the Distributive Property for this case, we can use it to solve for $\phi$ and $\phi^{-1}$:

$$\rho(x) = f(\phi^{-1}(x)) \tag{20}$$
$$\phi^{-1}(x) = f^{-1}(\rho(x)) \tag{21}$$
$$\phi^{-1}(x) = f^{-1}(e^x) \tag{22}$$

$$\rho^{-1}(x) = \phi(f^{-1}(x)) \tag{23}$$
$$\phi(x) = \rho^{-1}(f(x)) \tag{24}$$
$$\phi(x) = \log(f(x)) \tag{25}$$

Finally, substituting $\phi(x)$ and $\phi^{-1}(x)$ back into the equation for $\psi$, we get:

$$\psi(a, b) = \phi^{-1}\left(\phi(a) + \phi(b)\right) \tag{26}$$
$$\psi(a, b) = f^{-1}\left(e^{\log(f(a)) + \log(f(b))}\right) \tag{27}$$
$$\psi(a, b) = f^{-1}\left(e^{\log(f(a))} \cdot e^{\log(f(b))}\right) \tag{28}$$
$$\psi(a, b) = f^{-1}\left(f(a) \cdot f(b)\right) \tag{29}$$

$$\square$$

**Theorem A.3.** *For the special case of GenAgg parametrised by $\theta = \langle f, \alpha, \beta \rangle = \langle f, 0, 0 \rangle$, the Generalised Distributive Property for $\bigoplus_\theta$ is also satisfied by the binary operator $\psi(a, b) = f^{-1}(f(a) + f(b))$.*

*Proof.* From Lemma A.1, the Generalised Distributive Property is satisfied if:

$$\rho^{-1}\left(\frac{n^\alpha}{n} \sum \rho(c + x_i)\right) = \rho^{-1}\left(\frac{n^\alpha}{n} \sum \rho(x_i)\right) + c \tag{30}$$

$$\text{where } \rho(x) = f(\phi^{-1}(x)) \tag{31}$$

In this proof, we are given that $\alpha = 0$:

$$\rho^{-1}\left(\frac{1}{n} \sum \rho(c + x_i)\right) = \rho^{-1}\left(\frac{1}{n} \sum \rho(x_i)\right) + c \tag{32}$$

If we select $\rho(x) = x$, then we can show that the condition is satisfied:

$$\frac{1}{n}\sum(c + x_i) = \left(\frac{1}{n}\sum x_i\right) + c \tag{33}$$

$$\frac{1}{n}\sum x_i + \frac{1}{n}\sum c = \left(\frac{1}{n}\sum x_i\right) + c \tag{34}$$

$$\left(\frac{1}{n}\sum x_i\right) + c = \left(\frac{1}{n}\sum x_i\right) + c \tag{35}$$

Given that $\rho(x) = x$ satisfies the Distributive Property for this case, we can use it to solve for $\phi$ and $\phi^{-1}$:

$$\rho(x) = f(\phi^{-1}(x)) \tag{36}$$
$$\phi^{-1}(x) = f^{-1}(\rho(x)) \tag{37}$$
$$\phi^{-1}(x) = f^{-1}(x) \tag{38}$$

$$\rho^{-1}(x) = \phi(f^{-1}(x)) \tag{39}$$
$$\phi(x) = \rho^{-1}(f(x)) \tag{40}$$
$$\phi(x) = f(x) \tag{41}$$

Finally, substituting $\phi(x)$ and $\phi^{-1}(x)$ back into the equation for $\psi$, we get:

$$\psi(a, b) = \phi^{-1}\left(\phi(a) + \phi(b)\right) \tag{42}$$
$$\psi(a, b) = f^{-1}\left(f(a) + f(b)\right) \tag{43}$$

$\square$

# B Training Plots

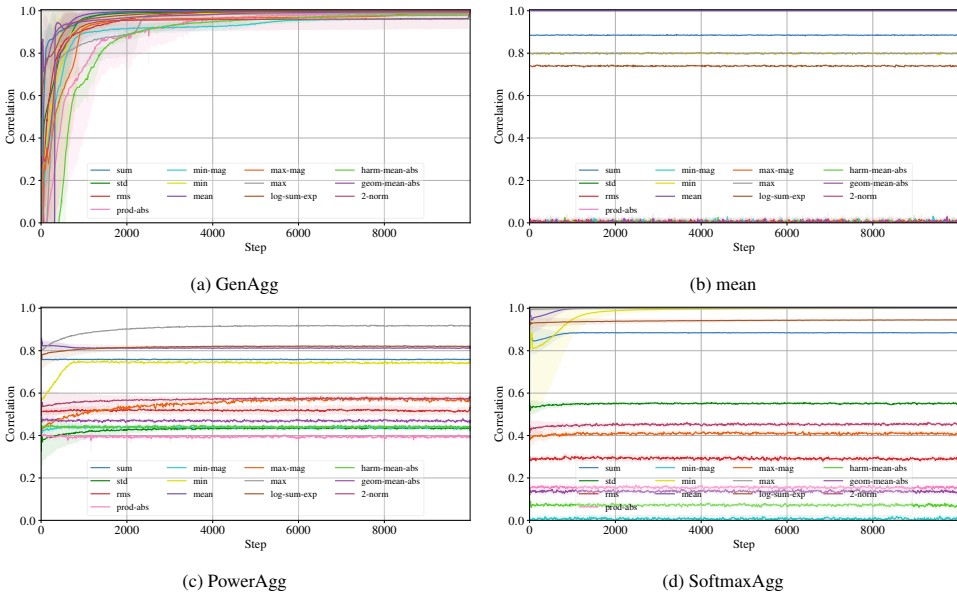

Figure 4: Training plots for the Aggregator Regression experiment (see Section 5.1). Each plot represents the ability of a parametrised aggregator $\bigoplus$ to regress over all standard aggregators $\bigodot_k \in \mathcal{A}$. The plots show the mean and standard deviation of the correlation between the predicted and ground truth values over 10 trials.

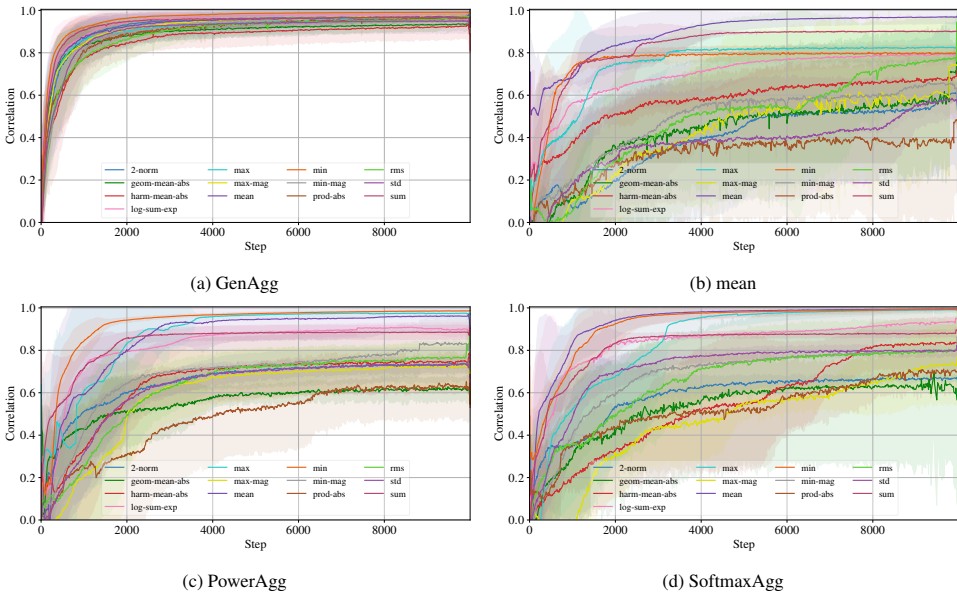

Figure 5: Training plots for the GNN Regression experiment (see Section 5.2). Each plot represents the ability of a GNN using parametrised aggregator $\bigoplus$ to regress over all standard aggregators $\bigodot_k \in \mathcal{A}$. The plots show the mean and standard deviation of the correlation between the predicted and ground truth values over 10 trials.

# C Parametrisations

In this section, we use the augmented $f$-mean to show that GenAgg is theoretically capable of representing all of the standard aggregators $\odot_k \in \mathcal{A}$. For each standard aggregator $\odot_k$, we prove that there exists a parametrisation $\theta$ such that $\bigoplus_\theta = \odot_k$. In a slight abuse of notation, mathematical operations applied to the input set $\mathcal{X}$ (such as an absolute value or a geometric inverse) denote elementwise operations: $f(\mathcal{X}) = \{f(x_1), \ldots, f(x_n)\}$.

**Theorem C.1.** *Mean. For $\theta = \langle f, \alpha, \beta \rangle = \langle x, 0, 0 \rangle$, the augmented $f$-mean equals the mean $\bigoplus_\theta = \frac{1}{n} \sum x_i$.*

*Proof.*

$$\bigoplus_{\langle x,0,0 \rangle \atop x_i \in \mathcal{X}} x_i = \left( n^{0-1} \sum_{x_i \in \mathcal{X}} (x_i - 0 \cdot \mu) \right) \tag{44}$$

$$= \frac{1}{n} \sum_{x_i \in \mathcal{X}} x_i \tag{45}$$

$\square$

**Theorem C.2.** *Sum. For $\theta = \langle f, \alpha, \beta \rangle = \langle x, 1, 0 \rangle$, the augmented $f$-mean equals the sum $\bigoplus_\theta = \sum x_i$.*

*Proof.*

$$\bigoplus_{\langle x,1,0 \rangle \atop x_i \in \mathcal{X}} x_i = \left( n^{1-1} \sum_{x_i \in \mathcal{X}} (x_i - 0 \cdot \mu) \right) \tag{46}$$

$$= \sum_{x_i \in \mathcal{X}} x_i \tag{47}$$

$\square$

**Theorem C.3.** *Product. For $\theta = \langle f, \alpha, \beta \rangle = \langle \log(|x|), 1, 0 \rangle$, the augmented $f$-mean equals the product $\bigoplus_\theta = \prod |x_i|$.*

*Proof.*

$$\bigoplus_{\langle \log(|x|),1,0 \rangle \atop x_i \in \mathcal{X}} x_i = e^{\left( n^{1-1} \sum_{x_i \in \mathcal{X}} \log(|x_i - 0 \cdot \mu|) \right)} \tag{48}$$

$$= e^{\sum_{x_i \in \mathcal{X}} \log(|x_i|)} \tag{49}$$

$$= \prod_{x_i \in \mathcal{X}} e^{\log(|x_i|)} \tag{50}$$

$$= \prod_{x_i \in \mathcal{X}} |x_i| \tag{51}$$

$\square$

**Theorem C.4.** *Max Magnitude. For $\theta = \langle f, \alpha, \beta \rangle = \langle \lim_{p \to \infty} |x|^p, 0, 0 \rangle$, the augmented $f$-mean equals the max magnitude $\bigoplus_\theta = \max(|\mathcal{X}|)$.*

*Proof.*

$$\bigoplus_{\langle \lim_{p \to \infty} |x|^p,0,0 \rangle \atop x_i \in \mathcal{X}} x_i = \lim_{p \to \infty} \left( n^{0-1} \sum_{x_i \in \mathcal{X}} |x_i - 0 \cdot \mu|^p \right)^{\frac{1}{p}} \tag{52}$$

$$= \lim_{p \to \infty} \frac{1}{n^{\frac{1}{p}}} \left( \sum_{x_i \in \mathcal{X}} |x_i|^p \right)^{\frac{1}{p}} \tag{53}$$

$$= \lim_{p \to \infty} \left( \sum_{x_i \in \mathcal{X}} |x_i|^p \right)^{\frac{1}{p}} \tag{54}$$

This aggregator is composed of monotonic functions, so if every element $x_i$ is substituted with $\max(|\mathcal{X}|)$, then the output should increase. Therefore, we can write the following inequality:

$$\lim_{p\to\infty}\left(\sum_{x_i\in\mathcal{X}}|x_i|^p\right)^{\frac{1}{p}} \leq \lim_{p\to\infty}\left(\sum_{i\in[1..n]}\max(|\mathcal{X}|)^p\right)^{\frac{1}{p}} \tag{55}$$

$$\lim_{p\to\infty}\left(\sum_{x_i\in\mathcal{X}}|x_i|^p\right)^{\frac{1}{p}} \leq \lim_{p\to\infty} n^{\frac{1}{p}}\cdot(\max(|\mathcal{X}|)^p)^{\frac{1}{p}} \tag{56}$$

$$\lim_{p\to\infty}\left(\sum_{x_i\in\mathcal{X}}|x_i|^p\right)^{\frac{1}{p}} \leq \max(|\mathcal{X}|) \tag{57}$$

$$\tag{58}$$

Similarly, since sum is monotonic, the value of the output computed over a set $\bigoplus_\theta(\mathcal{X})$ is greater than the output computed over a single element from that set $\bigoplus_\theta(\{x_i\})$, where $x_i \in \mathcal{X}$. Furthermore, we know that $\max(|\mathcal{X}|)$ is one of the elements of $\mathcal{X}$. So, we can write the following inequality:

$$\lim_{p\to\infty}\left(\sum_{x_i\in\mathcal{X}}|x_i|^p\right)^{\frac{1}{p}} \geq \lim_{p\to\infty}(\max(|\mathcal{X}|)^p)^{\frac{1}{p}} \tag{59}$$

$$\lim_{p\to\infty}\left(\sum_{x_i\in\mathcal{X}}|x_i|^p\right)^{\frac{1}{p}} \geq \max(|\mathcal{X}|) \tag{60}$$

$$\tag{61}$$

Consequently, by the squeeze theorem, we can state:

$$\lim_{p\to\infty}\left(\sum_{x_i\in\mathcal{X}}|x_i|^p\right)^{\frac{1}{p}} = \max(|\mathcal{X}|) \tag{62}$$

$\square$

**Theorem C.5.** *Min Magnitude*. *For* $\theta = \langle f,\alpha,\beta\rangle = \langle\lim_{p\to\infty}|x|^{-p},0,0\rangle$, *the augmented $f$-mean equals the min magnitude* $\bigoplus_\theta = \min(|\mathcal{X}|)$.

*Proof.*

$$\bigoplus_{\langle\lim_{p\to\infty}|x|^{-p},0,0\rangle \atop x_i\in\mathcal{X}} x_i = \lim_{p\to\infty}\left(n^{0-1}\sum_{x_i\in\mathcal{X}}|x_i - 0\cdot\mu|^{-p}\right)^{-\frac{1}{p}} \tag{63}$$

$$= \lim_{p\to\infty}\frac{1}{n^{-\frac{1}{p}}}\left(\sum_{x_i\in\mathcal{X}}|x_i|^{-p}\right)^{-\frac{1}{p}} \tag{64}$$

$$= \lim_{p\to\infty}\left(\sum_{x_i\in\mathcal{X}}|x_i|^{-p}\right)^{-\frac{1}{p}} \tag{65}$$

We use the theorem for the parametrisation of max magnitude (Theorem C.4) as a lemma for this proof. By applying a monotonically decreasing transformation to the inputs and then inverting that transformation on the output, we can write the min as a function of the max. Since the inputs are restricted to the positive domain with the absolute value, we select the transformation $T(x) = \frac{1}{x}$:

$$\min(|\mathcal{X}|) = \frac{1}{\max(\frac{1}{|\mathcal{X}|})} \tag{66}$$

Substituting the parametrisation from Theorem C.4 for max, we get:

$$\min(|\mathcal{X}|) = \frac{1}{\lim_{p \to \infty} \left( \sum_{x_i \in \mathcal{X}} \left( \frac{1}{|x_i|} \right)^p \right)^{\frac{1}{p}}} \tag{67}$$

$$= \lim_{p \to \infty} \left( \sum_{x_i \in \mathcal{X}} |x_i|^{-p} \right)^{-\frac{1}{p}} \tag{68}$$

$\square$

**Theorem C.6. *Max*.** *For* $\theta = \langle f, \alpha, \beta \rangle = \langle \lim_{p \to \infty} e^{px}, 0, 0 \rangle$, *the augmented $f$-mean equals the max:* $\bigoplus_\theta = \max(\mathcal{X})$.

*Proof.*

$$\bigoplus_{\langle \lim_{p \to \infty} e^{px}, 0, 0 \rangle} x_i = \lim_{p \to \infty} \frac{1}{p} \log \left( n^{0-1} \sum_{x_i \in \mathcal{X}} e^{p(x_i - 0 \cdot \mu)} \right) \tag{69}$$

$$= \lim_{p \to \infty} \frac{1}{p} \log \left( \frac{1}{n} \right) + \frac{1}{p} \log \left( \sum_{x_i \in \mathcal{X}} e^{p \cdot x_i} \right) \tag{70}$$

$$= \lim_{p \to \infty} \log \left( \left( \sum_{x_i \in \mathcal{X}} e^{p \cdot x_i} \right)^{\frac{1}{p}} \right) \tag{71}$$

$$\tag{72}$$

This aggregator is composed of monotonic functions, so if every element $x_i$ is substituted with $\max(\mathcal{X})$, then the output should increase. Therefore, we can write the following inequality:

$$\lim_{p \to \infty} \log \left( \left( \sum_{x_i \in \mathcal{X}} e^{p \cdot x_i} \right)^{\frac{1}{p}} \right) \leq \lim_{p \to \infty} \log \left( \left( \sum_{x_i \in \mathcal{X}} e^{p \cdot \max(\mathcal{X})} \right)^{\frac{1}{p}} \right) \tag{73}$$

$$\leq \lim_{p \to \infty} \log \left( n^{\frac{1}{p}} \cdot \left( e^{p \cdot \max(\mathcal{X})} \right)^{\frac{1}{p}} \right) \tag{74}$$

$$\leq \log \left( e^{\max(\mathcal{X})} \right) \tag{75}$$

$$\leq \max(\mathcal{X}) \tag{76}$$

$$\tag{77}$$

Similarly, since sum is monotonic, the value of the output computed over a set $\bigoplus_\theta(\mathcal{X})$ is greater than the output computed over a single element from that set $\bigoplus_\theta(\{x_i\})$, where $x_i \in \mathcal{X}$. Furthermore, we know that $\max(\mathcal{X})$ is one of the elements of $\mathcal{X}$. So, we can write the following inequality:

$$\lim_{p \to \infty} \log \left( \left( \sum_{x_i \in \mathcal{X}} e^{p \cdot x_i} \right)^{\frac{1}{p}} \right) \geq \lim_{p \to \infty} \log \left( \left( e^{p \cdot \max(\mathcal{X})} \right)^{\frac{1}{p}} \right) \tag{78}$$

$$\geq \log \left( e^{\max(\mathcal{X})} \right) \tag{79}$$

$$\geq \max(\mathcal{X}) \tag{80}$$

$$\tag{81}$$

Consequently, by the squeeze theorem, we can state:

$$\lim_{p \to \infty} \log \left( \left( \sum_{x_i \in \mathcal{X}} e^{p \cdot x_i} \right)^{\frac{1}{p}} \right) = \max(\mathcal{X}) \tag{82}$$

$\square$

**Theorem C.7. Min.** *For $\theta = \langle f, \alpha, \beta \rangle = \langle \lim_{p \to \infty} e^{-px}, 0, 0 \rangle$, the augmented $f$-mean equals the min:* $\bigoplus_{\theta} = \min(\mathcal{X})$.

*Proof.*

$$\bigoplus_{\substack{\langle \lim_{p \to \infty} e^{-px}, 0, 0 \rangle \\ x_i \in \mathcal{X}}} x_i = \lim_{p \to \infty} -\frac{1}{p} \log \left( n^{0-1} \sum_{x_i \in \mathcal{X}} e^{-p(x_i - 0 \cdot \mu)} \right) \tag{83}$$

$$= \lim_{p \to \infty} -\frac{1}{p} \log \left( \frac{1}{n} \right) - \frac{1}{p} \log \left( \sum_{x_i \in \mathcal{X}} e^{-p \cdot x_i} \right) \tag{84}$$

$$= \lim_{p \to \infty} \log \left( \left( \sum_{x_i \in \mathcal{X}} e^{-p \cdot x_i} \right)^{-\frac{1}{p}} \right) \tag{85}$$

$$\tag{86}$$

We use the theorem for the parametrisation of max (Theorem C.6) as a lemma for this proof. By applying a monotonically decreasing transformation to the inputs and then inverting that transformation on the output, we can write the min as a function of the max. We select the transformation $T(x) = -x$:

$$\min(\mathcal{X}) = -\max(-\mathcal{X}) \tag{87}$$

Substituting the parametrisation from Theorem C.6 for max, we get:

$$\min(\mathcal{X}) = \lim_{p \to \infty} -\log \left( \left( \sum_{x_i \in \mathcal{X}} e^{p \cdot (-x_i)} \right)^{\frac{1}{p}} \right) \tag{88}$$

$$= \lim_{p \to \infty} \log \left( \left( \sum_{x_i \in \mathcal{X}} e^{-p \cdot x_i} \right)^{-\frac{1}{p}} \right) \tag{89}$$

$\square$

**Theorem C.8. Harmonic Mean.** *For $\theta = \langle f, \alpha, \beta \rangle = \langle \frac{1}{x}, 0, 0 \rangle$, the augmented $f$-mean equals the harmonic mean* $\bigoplus_{\theta} = \frac{n}{\sum \frac{1}{x_i}}$.

*Proof.*

$$\bigoplus_{\substack{\langle x, 0, 0 \rangle \\ x_i \in \mathcal{X}}} x_i = \left( n^{0-1} \sum_{x_i \in \mathcal{X}} (x_i - 0 \cdot \mu)^{-1} \right)^{-1} \tag{90}$$

$$= \left( \frac{1}{n} \sum_{x_i \in \mathcal{X}} \frac{1}{x_i} \right)^{-1} \tag{91}$$

$$= \frac{n}{\sum_{x_i \in \mathcal{X}} \frac{1}{x_i}} \tag{92}$$

$\square$

**Theorem C.9.** *Geometric Mean. For $\theta = \langle f, \alpha, \beta \rangle = \langle \log(|x|), 0, 0 \rangle$, the augmented $f$-mean equals the geometric mean $\bigoplus_\theta = \sqrt[n]{\prod |x_i|}$.*

*Proof.*

$$\bigoplus_{\substack{x_i \in \mathcal{X}}}{}_{\langle \log(|x|), 0, 0 \rangle} x_i = e^{\left( n^{0-1} \sum_{x_i \in \mathcal{X}} \log(|x_i - 0 \cdot \mu|) \right)} \tag{93}$$

$$= e^{\frac{1}{n} \sum_{x_i \in \mathcal{X}} \log(|x_i|)} \tag{94}$$

$$= \prod_{x_i \in \mathcal{X}} e^{\frac{1}{n} \log(|x_i|)} \tag{95}$$

$$= \prod_{x_i \in \mathcal{X}} e^{\log(|x_i|^{\frac{1}{n}})} \tag{96}$$

$$= \prod_{x_i \in \mathcal{X}} |x_i|^{\frac{1}{n}} \tag{97}$$

$$= \sqrt[n]{\prod_{x_i \in \mathcal{X}} |x_i|} \tag{98}$$

$\square$

**Theorem C.10.** *Root Mean Square. For $\theta = \langle f, \alpha, \beta \rangle = \langle x^2, 0, 0 \rangle$, the augmented $f$-mean equals the root mean square $\bigoplus_\theta = \sqrt{\frac{1}{n} \sum x_i^2}$.*

*Proof.*

$$\bigoplus_{\substack{x_i \in \mathcal{X}}}{}_{\langle x^2, 0, 0 \rangle} x_i = \left( n^{0-1} \sum_{x_i \in \mathcal{X}} (x_i - 0 \cdot \mu)^2 \right)^{\frac{1}{2}} \tag{99}$$

$$= \sqrt{\frac{1}{n} \sum_{x_i \in \mathcal{X}} x_i^2} \tag{100}$$

$$\tag{101}$$

$\square$

**Theorem C.11.** *Euclidean Norm. For $\theta = \langle f, \alpha, \beta \rangle = \langle x^2, 1, 0 \rangle$, the augmented $f$-mean equals the euclidean norm $\bigoplus_\theta = \sqrt{\sum x_i^2}$.*

*Proof.*

$$\bigoplus_{\substack{x_i \in \mathcal{X}}}{}_{\langle x^2, 1, 0 \rangle} x_i = \left( n^{1-1} \sum_{x_i \in \mathcal{X}} (x_i - 0 \cdot \mu)^2 \right)^{\frac{1}{2}} \tag{102}$$

$$= \sqrt{\sum_{x_i \in \mathcal{X}} x_i^2} \tag{103}$$

$$\tag{104}$$

$\square$

**Theorem C.12.** *Standard Deviation. For $\theta = \langle f, \alpha, \beta \rangle = \langle x^2, 0, 1 \rangle$, the augmented $f$-mean equals the standard deviation $\bigoplus_\theta = \sqrt{\sum (x_i - \mu)^2}$.*

*Proof.*

$$\bigoplus_{\substack{x_i \in \mathcal{X}}}{}_{\langle x^2, 0, 1 \rangle} x_i = \left( n^{0-1} \sum_{x_i \in \mathcal{X}} (x_i - 1 \cdot \mu)^2 \right)^{\frac{1}{2}} \tag{105}$$

$$= \sqrt{\frac{1}{n} \sum_{x_i \in \mathcal{X}} (x_i - \mu)^2} \tag{106}$$

$$\tag{107}$$

□

**Theorem C.13.** *Log-Sum-Exp. For $\theta = \langle f, \alpha, \beta \rangle = \langle e^x, 0, 1 \rangle$, the augmented $f$-mean equals the log-sum-exp* $\bigoplus_\theta = \log(\sum e^{x_i})$.

*Proof.*

$$\bigoplus_{\langle e^x, 1, 0 \rangle \atop x_i \in \mathcal{X}} x_i = \log \left( n^{1-1} \sum_{x_i \in \mathcal{X}} e^{x_i - 0 \cdot \mu} \right) \tag{108}$$

$$= \log \left( \sum_{x_i \in \mathcal{X}} e^{x_i} \right) \tag{109}$$

$$\tag{110}$$

□

# D   Limitations

In our problem statement, we define a set of special cases $\mathcal{A}$ which we refer to as "standard aggregators". Our regression experiments analyse the representational capacity of various methods by analysing their ability to regress over the aggregators in $\mathcal{A}$. However, we acknowledge that this set is not exhaustive, so there may exist special cases not in $\mathcal{A}$ which are useful or could provide some additional insight.

Our GNN benchmark experiments also only provide data about the performance of GenAgg in a limited number of applications. We evaluate on the GNN benchmark datasets suite from [6], which includes MNIST, CIFAR10, CLUSTER, and PATTERN. These datasets provide a mix of node classification and graph classification tasks (to complement the regression tasks from our other experiments). We did not include the TSP and CSL datasets from the same GNN benchmarks suite because TSP is an edge classification task (which would necessitate significant modification of our GNN), and CSL requires node positional encodings in order to be solvable by message passing GNNs, which it does not include by default [6]. In our initial testing, we also considered using the PUBMED, CORA, and CITESEER datasets. However, they are extremely small datasets that often lead to overfitting. Furthermore, there is something fundamental about the coauthor problem (upon which all three datasets are based) that fundamentally does not require the same level of complexity to solve—the train accuracy of all methods, including simple aggregators, approaches 1. Instead of these coauthor datasets, we opted to use the GNN benchmark dataset, which seemed to present more "difficult" problems. However, for the sake of transparency, we include our results on the datasets that we decided not to use:

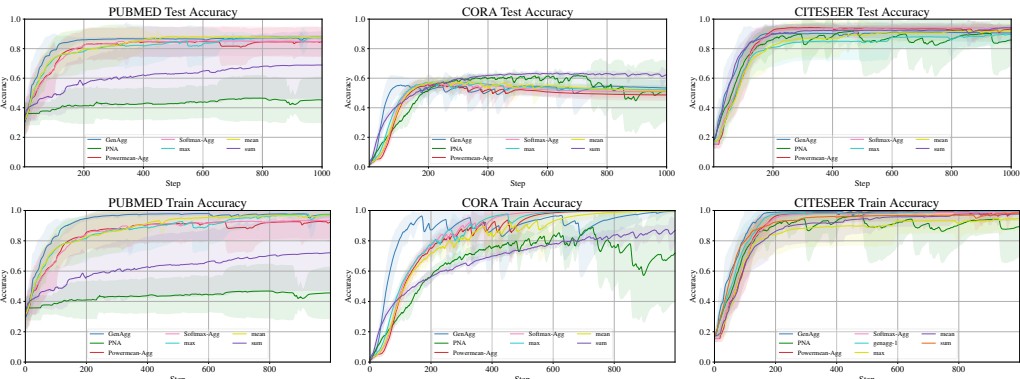

Figure 6: Train and Test Accuracy for all baselines on PUBMED, CORA, and CITESEER.

In these small datasets, the primary problem across all models is overfitting. GenAgg performs on-par with the best baseline, but it does not exhibit a performance *boost* as it does in the other datasets. To determine if the lack of improvement is due to the overfitting or the dataset itself, we run an additional experiment that explicitly examines the effect of overfitting by artificially reducing the amount of training data (Figure 7):

|      | GenAgg | Best Baseline | Median Baseline |      | GenAgg | Best Baseline | Median Baseline |
|------|--------|---------------|-----------------|------|--------|---------------|-----------------|
| 100% | **0.915** | 0.872 | 0.841 | 100% | **0.926** | 0.897 | 0.866 |
| 10%  | **0.958** | 0.903 | 0.846 | 10%  | **0.905** | 0.854 | 0.829 |
| 1%   | **0.978** | 0.846 | 0.832 | 1%   | **0.903** | 0.832 | 0.830 |
| (a) Train Accuracy | | | | (b) Test Accuracy | | | |

Figure 7: Train and test accuracy of GenAgg vs all baselines on various subsets of the PATTERN dataset. We train on the PATTERN dataset (100% of the data), and versions with 10% and 1% of the original datapoints.

This experiment highlights a difference between the small coauthor datasets and the reduced PATTERN dataset (Figure 7). In the coauthor datasets the train accuracies approach 1, whereas in the reduced PATTERN dataset the baselines do not surpass a certain level of performance. This indicates that the mathematical relationships in the PATTERN dataset are fundamentally more difficult to represent. It is in these more complex problems that GenAgg provides the most benefit.

# E    Training Details

In our implementation of GenAgg, we implement $f$ and $f^{-1}$ as MLPs with hidden sizes of $[1, 2, 2, 4]$ and $[4, 2, 2, 1]$ using Mish activation, BatchNorm, and Kaiming Normal weight initialisation. To run the GNN benchmark experiments, we use a 4-layer GraphConv model with a hidden size of 64, using Mish activation between layers. MLPs are used as pre- and post- processors in order to map to and from the hidden dimension of the GNN. The preprocessor is implemented with a one layer MLP, and the postprocessor is implemented with a 4-layer MLP using Mish activation. In tasks which require graph-level predictions, we prepend a global mean pooling layer to the postprocessor.

We run all of our experiments on an NVIDIA GeForce GTX 1080 Ti GPU. In all experiments, we use the Adam optimiser with a learning rate of $10^{-3}$. In the regression experiments we train for $10, 000$ epochs with a batch size of 1024, and in the GNN benchmark experiment we train for $1, 000$ epochs with a batch size of 32. Our results report the mean (and standard deviation, as the shaded region in the training plots) over 10 trials.

# F    Runtime

As GenAgg requires running forward passes of small neural networks in addition to performing a sum, it incurs an additional runtime cost. We report the runtime overhead of each method in the figure below:

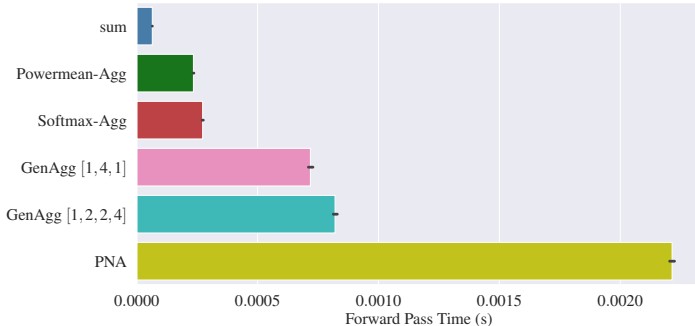

Figure 8: Forward pass times for each aggregation method. GenAgg $[1, 2, 2, 4]$ and GenAgg $[1, 4, 1]$ denote two different layer architectures for $f$. The data in this figure represents the time for a single forward pass over the MNIST GNN Benchmark dataset with a batch size of 1024 (approximately 578k edges), using an NVIDIA GeForce GTX 1080 Ti GPU. The reported time is *only* for the aggregation component, not the GNN as a whole.

While the the absolute runtime of GenAgg is relatively fast, it is still significantly slower than sum. Consequently, it is possible that the runtime of our implementation can preclude its use in time-critical applications. However, note that this figure only represents the runtime for our specific implementation—GenAgg can be implemented with any invertible function $f$. Using a symbolic parametrisation of invertible functions can significantly speed up computations (at the cost of representational complexity). Alternatively, it is likely that an implementation in JAX with compiled networks $f$ and $f^{-1}$ can achieve a speed boost with the same architecture.