# OpenReview forum: "Generalised f-Mean Aggregation for Graph Neural Networks"
_NeurIPS.cc/2023/Conference — NeurIPS 2023 poster_

### Official Review · Reviewer_PwWR · 2023-07-03

**Soundness:** 3 good
**Presentation:** 3 good
**Contribution:** 3 good
**Rating:** 7
**Confidence:** 4

**Summary:**

The authors deal with the interesting problem of parametrising the space of aggregation functions. They propose a novel aggregation method that allows each MPNN to learn the most appropriate aggregator for each downstream task. In contrast to previous works, their proposed aggregator can interpret a broad pool of known aggregators and enjoys a set of theoretical observations that can guarantee the generalization of the distributive property.

The proposed aggregator is based on the generalized f-mean, a known operator that imposes though a set of restrictive constraints with respect symmetry, monotonicity, and idempotency. For that reason, they extend the definition of the generalized f-mean operator with parameters $\alpha$ and $\beta$ that relax the two latter constraints, while maintaining the symmetry property.

Using two separate MLPs, they implement the instances of functions $f$ and $f^{-1}$ for the definition of the augmented f-mean operator, and they propose a new optimization objective (Equation 2), in order to ensure the equality of the invertability of function $f$ to the original input signal $x$.

Their extensive experimentation study suggests the superior performance of their proposed aggregator (GenAgg) in both computer vision and graph downstream tasks. Also, Theorem 3.1 suggests that GenAgg satisfies a generalised distributive property, allowing for further analysis of the algorithmic time  and space complexity of common aggregators.



**Strengths:**

- The idea of extending the generalized f-mean operator for defining a new aggregation framework is novel, and provides insights on the effectiveness of common aggregators in MPNNs.
- GenAgg can be the framework for defining already defined aggregation operators, as well as enabling the design of new, more sophisticated ones.
- Theorem 3.1 allows for a further analysis of common aggregators with respect to their time and space complexity.
- The empirical results suggest a very strong performance of GenAgg with respect to other powerful MPNN aggregators.

**Weaknesses:**

- There is not a clear connection of the evolution of parameters $\alpha$ and $\beta$ with respect to the downstream tasks. Especially, when the datasets come from different domains (e.g computer vision, and graph data), it is unclear how the GenAgg is formed, and whether the final configuration correspond to an optimal representation of the neighborhood adjacency.
- It would be very helpful for the authors to discuss the convergence analysis of the loss term defined in Equation 2. It seems from the results in the computer vision datasets, that the model is quickly converged to a state-of-the-art performance. Is that pattern observed, also, in the GNN benchmark datasets?

**Questions:**

1. There is a relation of the proposed parameterization of the aggregation operators with a couple of other proposed parametrised operators [1],[2]. Can the authors discuss any potential advantage over these methods, and whether there is a theoretical or intuitive connection among them?


[1] Dan Zhang et al. Node-Wise Adaptive Aggregation in GNNs for Recommendation. 2023
[2] George Dasoulas et al. Learning Parametrised Graph Shift Operators. 2021


**Limitations:**

The authors discuss potential limitations of their method.

---

> ### Author Rebuttal · Authors · 2023-08-07
>
> Thank you for your review! We will address the weaknesses that you cited:
>
> - If I understand this correctly, this is about the explainability of GenAgg—is that correct? You are correct, some datasets (such as images, as you mentioned) are less likely to lead to an easily interpretable aggregator. Any learned aggregator can be analysed to some degree (e.g. the dependence on the degree, the relative importance of high/low values, the dependence on the variation of the data) by the method outlined in our discussion section, but the aggregator will not always converge to a satisfying special case. To maximise the explainability properties of GenAgg, we would suggest using it in a proxy model with data which is likely to have some symbolic form (e.g. geometric data, physics-based data, etc).
> - This bullet brings up two points. To answer the first: the loss term from Equation 2 converges orders of magnitude faster than the model as a whole. In effect, $f$ and $f^{-1}$ remain true inverses of each other throughout training, and evolve simultaneously as they converge to the desired parametrisation. To answer the second point: All four of the datasets in this paper are GNN datasets, two of which are adapted from computer vision data, and two of which are based on mathematical modelling tasks.
>
> To answer your question about the the relationship between GenAgg and the methods you have cited:
>
> - *Node-Wise Adaptive Aggregation in GNNs for Recommendation*.
> This paper is about a new GNN architecture for the purpose of learning on heterogeneous graphs. This is achieved by conditioning the weights on metrics which can distinguish the nodes from each other, such as the degree. This focuses on a different part of the GNN—while GenAgg parametrises the aggregation function, this method parametrises the weights applied to each node. To avoid confusion, note that this paper defines aggregation slightly differently than we do. For example, the paper regards attention as a form of aggregation, while we would consider it a mechanism which *contains* a sum aggregator.
> - *Learning Parametrised Graph Shift Operators*.
> This method parametrises graph shift operators, generalising special cases like the adjacency, the normalised form of the adjacency used in GCN, and the laplacian. While GenAgg focuses on just aggregation, this method sets the edge weights which are then used in the aggregation step (the aggregation they use is a sum, so it is always linear regardless of the GSO parametrisation). In this sense, the methods are complementary—one could use both approaches at the same time. The one aspect of overlap is normalisation based on the degree. While GenAgg bases the normalisation exclusively on the degree of node $i$, this method can also normalise by a combination of the degrees of node $i$ and node $j$.

---

> > ### Comment · Reviewer_PwWR · 2023-08-16
> >
> > Thank you for your reply!
> > You addressed my concerns satisfyingly, so I will retain my score to Accept (7).

---

### Official Review · Reviewer_uMEa · 2023-07-08

**Soundness:** 3 good
**Presentation:** 3 good
**Contribution:** 2 fair
**Rating:** 5
**Confidence:** 4

**Summary:**

This paper presents GenAgg, a generalised aggregation operator, which parametrises a function space that includes all standard aggregators. They show using GenAgg as a drop-in replacement for an existing aggregator in a GNN usually leads to a boost in performance across various tasks.

**Strengths:**

1. The proposed GenAgg is an unified aggregator that parametrizes a function space, encompassing all standard aggregators.

2. Using GenAgg as a drop-in replacement for the aggregators in a GNN often leads to a performance boost.

3. The paper is well-structured. Table 1 is particularly commendable as it lucidly demonstrates that the proposed GenAgg can accommodate most common aggregators.

**Weaknesses:**


1. The tradeoff is not mentioned. GenAgg requires to uses 8 additional linear layers (4 layers MLPs for f and f inverse). Despite the channel size of each layer is small, it will still lower down inference speed significantly. The latency of using different aggregators should be included in Figure 3 Table. The fair comparison for different aggregators in the same latency (by changing the number of channels) should be considered as well.

2. The authors should consider providing GenAgg with various baselines. Currently, the demonstration of GenAgg is limited to its use with a simple 4-layer GraphConv model. It would be insightful to see experiments using more powerful baselines such as GraphSAGE, DeepGCNs, and state-of-the-art networks. There's a possibility that as network capability increases, the selection of aggregation functions diminishes in significance. This brings into question another aspect of this paper: its practicality.

3. The practical application of GenAgg appears to be limited for large networks. Given that GenAgg slows down the processing speed, it prompts the question: Is it essential to implement GenAgg in a deep network? It would be beneficial if the authors could demonstrate at least one instance of its application in a larger graph (for example, OGB benchmarks) using a larger network (over 50 layers), rather than limiting their illustrations to small networks in toy examples.


**Questions:**

see weakness.

**Limitations:**

The authors mentioned overfitting cases in their limitation part. I also point out the latency issues, I would like to know the feedback from authors.

---

> ### Author Rebuttal · Authors · 2023-08-07
>
> Thank you for the review! To address the weaknesses:
>
> - You’re right, GenAgg does introduce some computational overhead compared to an aggregator like ‘sum’. Unfortunately, an experiment where we tweak the architectures until all achieve the same latency is not possible—sum is a subcomponent of GenAgg, so it will always be less computationally expensive. However, we note that a forward pass takes less than 1ms (using MNIST data, processing a batch containing over 500,000 edges). Given the scale of this runtime, the difference in runtime for larger architectures using GenAgg vs other aggregators as subcomponents should be relatively small. That being said, the focus of this paper is not on runtime, but on performance. - We examine a different tradeoff: representational capacity vs overfitting (to address this, we report test accuracy, and we discuss the tradeoff with the number of model parameters). However, we agree that latency is useful information for some, so we will include a table of the latencies in the Limitations section (see annexed PDF), as well as a discussion about the results.
> - Ideally we would like to show that GenAgg works with all different GNN architectures and numbers of layers. However, due to the impracticality of running the same experiments for multiple trials on several different setups, we chose to select a representative architecture. To maximise the relevance of the results, we selected GraphConv (a popular architecture), and chose the number of layers to be 4 in accordance with the details in the Benchmarking GNNs paper [1]. As an aside, we note that the Pytorch Geometric implementation of SAGEConv is equivalent to GraphConv with a mean aggregator instead of sum (the project and normalise operations are disabled by default), so implicitly we did test that particular architecture. To address your concern about the usefulness of aggregators as the number of layers increases: you’re right that it is possible that the benefit of aggregators might see diminishing returns after the first layer. However, it is likely that the aggregation in the first layer will remain extremely important, as a naive aggregation can irreversibly remove important information from the raw input (for example, if we wish to predict the standard deviation in the distances of neighbouring agents from their positions, then if we apply a sum in the first layer, it is not possible to reconstruct the correct output, no matter how many layers there are). To some degree, this result can be seen in the second experiment: even as we introduce more than enough representational complexity in a GNN, it is unable to overcome a poor choice in aggregator.
> - Most of this question was addressed in the previous answer, but we will briefly elaborate. We would not consider the GNN Benchmark experiment a “small network in a toy example”. The Pattern and Cluster datasets include 10,000 graphs, each of which contain an average of 4,000-6,000 edges. To maximise relevance to the community, we strove to select the most popular datasets and use the most popular architecture (we note that the models in the papers associated with the datasets on pytorch geometric often use ~4 layers). However, we agree that in the future it would be interesting to test GenAgg with different architectures.
>
> [1] Vijay Prakash Dwivedi, Chaitanya K Joshi, Thomas Laurent, Yoshua Bengio, and Xavier Bresson. Benchmarking graph neural networks. arXiv preprint arXiv:2003.00982, 2020.

---

> > ### Comment · Reviewer_uMEa · 2023-08-15
> > **Thanks for the reply**
> >
> > I keep my original score as I did not see authors to conduct experiments using a GNN over 50 layers on a really large graph (over 100K nodes). All analysis of the paper is limited to networks of less than 5 layers and graphs with nodes less than 10K. It is not clear to me whether the propose method will be effective in large-scale graphs/networks or not.

---

> > > ### Author Response · Authors · 2023-08-15
> > > **Reply to Reviewer**
> > >
> > > The datasets that we tested have significantly more than 10k nodes. They contain from 10,000 to 55,000 graphs, each of which contain ~100 nodes (as opposed to most "larger" datasets, which only contain a single graph). In total, they range from 1.2 million to 5.3 million nodes. Are you primarily concerned with testing on a dataset where all of the nodes are in the _same graph_? Note that when training, the batch dimension is combined into a single graph anyway (furthermore, the full graph is not considered all at once for large single graph datasets---small subgraphs are sampled during training).
> > >
> > > With respect to the number of layers, we mentioned in our previous comment that we followed the architectures that we found in literature (as the scope of this paper concerns evaluating a new aggregator, not developing new extremely deep GNNs). However, we also tested with 16 layers instead of 4, and we found the the test accuracy decreased (regardless of which aggregator was used).

---

> > > > ### Comment · Reviewer_uMEa · 2023-08-17
> > > >
> > > > The reasons why I ask for large-scale experiment using deep GCNs are that: (1) the trend of the AI filed is to use deep networks; (2) it is not clear whether aggregation matters at all in the large-scale setting or not; (3) it is not clear whether your methods are bottlenecked in shallow networks. For example, whether the proposed MLPs can be optimized to learn a good aggregation function or not and whether it will siginpificantly impact efficiency in deep GCNs. Also, I am also curious why not just compare with the GenAgg in DeeperGCN using the same 100 layers.

---

> > > > > ### Author Response · Authors · 2023-08-19
> > > > > **New Experimental Results**
> > > > >
> > > > > We have run the additional experiment that you requested, using the OGB datasets with the DeeperGCN architecture. The primary bottleneck for this experiment was limited computational resources, particularly for such a large model and dataset in a short time frame. Consequently, we have had to run this experiment with reduced batch/subgraph sizes compared to the original paper (although all methods and baselines use the same settings), and we have only completed one trial so far. We will update the results as they come in, but most of the relevant results are present. We will continue running this experiment after the review period to collect more results, and add those results to the appendix.
> > > > >
> > > > > **ogbg-molhiv**
> > > > > |          | Sum    | SoftmaxAgg | GenAgg  |
> > > > > |----------|--------|------------|---------|
> > > > > | 3 Layer  | 0.7834 | 0.7901    | **0.7902**  |
> > > > > | 16 Layer | 0.7724 | 0.7955    | **0.8048**  |
> > > > > | 50 Layer | 0.7941 | running    | running |
> > > > >
> > > > >
> > > > > **ogbn-proteins**
> > > > > |          | Sum    | SoftmaxAgg | GenAgg |
> > > > > |----------|--------|------------|--------|
> > > > > | 3 Layer  | 0.8188 | **0.8201**     | 0.8195 |
> > > > > | 16 Layer | 0.8213 | 0.8222     | **0.8228** |
> > > > > | 50 Layer | **0.8254** | 0.8227    | **0.8254** |
> > > > >
> > > > > In general, GenAgg provides a marginal increase in performance. The margin of the performance gap seems to be more a function of the dataset than the number of layers---the performance increase in the molhiv dataset is larger than that of the proteins dataset. This aligns with the findings from the new experiment for Reviewer rgaT, which points to the conclusion that some datasets inherently require more complex mathematical aggregations. In the future, it would be interesting to test GenAgg in a deep GNN in a such a dataset---it could serve as a platform for ablation, to answer if GenAgg is required in *every* layer, or if it is sufficient to use it in only the first layer to avoid irreversibly losing information with a naive aggregation.

---

> > > > > > ### Comment · Reviewer_uMEa · 2023-08-21
> > > > > >
> > > > > > Dear authors, thanks for providing this experiment. First, please finish all experiments and consider to put them in your revision. Studying the effects of aggregation in deep and large-scale cases are important.
> > > > > > Second, your experiments can not address my concern of the necessities in studying aggregation functions in large-scale cases since the improvements as you said are very marginal to the sum aggregator. The proposed GenAgg also only shows neglectable improvements over the baseline (SoftmaxAgg).
> > > > > > Last, seems you forgot to provide the cost of latency of GenAgg in these large-scale experiments, could you add them?
> > > > > > Thank you.

---

> > > > > ### Comment · Area_Chair_Hv8M · 2023-08-21
> > > > > **Do the new results change your evaluation?**
> > > > >
> > > > > Dear reviewer uMEa,
> > > > >
> > > > > do the newly added results change your evaluation?
> > > > >
> > > > > Best!
> > > > > AC

---

> > > > > > ### Comment · Reviewer_uMEa · 2023-08-21
> > > > > >
> > > > > > I still have concerns:
> > > > > > 1) cost of efficiency (number of parameters, latency). The proposed aggregation has to use additional MLPs just to learn the aggregation functions.  This strategy is expensive and not sure the improvement is from the better aggregation or the increased parameters.
> > > > > > 2) improvement of the proposed aggregations is marginal (less than 0.1%) in large-scale experiments. This means the aggregation function is not very practical in some large-scale experiments.
> > > > > >
> > > > > > Theoretically, the proposed aggregation can approach any aggregator functions due to the universal approximation capability of MLPs. Therefore, this is a theoretically sound paper, but may not be very practical. Some important questions are not clearly addressed:
> > > > > > 1) how important is the aggregation to the network especially when the network is large (the authors failed to prove it is important, and somehow expected, remember convolutions just use sum as aggregation in 2D filed)
> > > > > >
> > > > > > 2) how expensive the proposed methods is (theoretically and experimentally). The authors said it is not expensive since it is only some MLPs, but the proposed methods requires more layers of convolutions that leads to slow speed. I suggest to study this carefully
> > > > > >
> > > > > > 3) how hard to optimize MLPs to learn useful aggregations?
> > > > > >
> > > > > > Anyway, I vote for borderline reject for now.

---

> > > > > > > ### Author Response · Authors · 2023-08-21
> > > > > > > **Reply to Reviewer uMEa**
> > > > > > >
> > > > > > > We will continue running the experiments and update the paper. The latency results for each aggregator can be found in the pdf included with our rebuttal. We will attempt to address your concerns:
> > > > > > >
> > > > > > > 1. All of the parameters in GenAgg are used for parameterising a larger class of aggregation functions (i.e. they provide an orthogonal form of representational capacity than that of the parameters in the convolutions). Of course, more parameters are required to represent more aggregation functions. We acknowledge that GenAgg is slower than taking a sum---when prioritising for runtime, sum is likely the best choice. However, in this paper we prioritise maximising validation performance. The number of parameters is much less important to performance than the architecture---for example, as we wrote to reviewer 3j4u, PNA has over 49,000 parameters for a feature dimension of 64, while GenAgg has less than 50. Nevertheless, GenAgg outperforms PNA through its ability to represent any standard aggregator and its use of constraints to accelerate learning.
> > > > > > >
> > > > > > > 2. While the performance increase in the dataset and model (OGB, DeeperGCN) that you suggested is marginal, we note that the performance difference between _all_ methods on that dataset is marginal. Increasing the number of layers from 3 to 50, or using the aggregator proposed in DeeperGCN (SoftmaxAgg) also have extremely marginal effects. It is notable that there is any performance increase at all over the paper's proposed model by using GenAgg with zero hyperparameter tuning. Note that GenAgg is not guaranteed to produce better performance on every dataset---it only helps in datasets where the default aggregator (e.g. sum) is not optimal. As we have seen in the experiment for reviewer rgaT, the performance boost is highly dependent on the dataset, providing a benefit on datasets which require mathematically more complex aggregations. Despite the fact that OGB is not such a dataset (as we can see from the very marginal improvement of the paper's proposed SoftmaxAgg over sum), it is comforting to see that GenAgg matches or improves upon the performance of the best baseline.
> > > > > > >
> > > > > > > And your questions:
> > > > > > >
> > > > > > > 1. As aggregation is a lossy operation, theoretically a learnable aggregation function is very important in the first layer of a network regardless of how large it is, as a poor choice in aggregation function can irreversibly lose important information. Although a thorough analysis of aggregation in extremely deep GNNs is outside the scope of this paper, it would be an interesting avenue for future research. In our paper, we have demonstrated that GenAgg is useful on the GNN architectures which are used the vast majority of the time (4-layers).
> > > > > > >
> > > > > > > 2. We have provided an analysis of the forward pass times for each aggregation method. The second part of this question is somewhat unclear--there are no convolutions inside of GenAgg.
> > > > > > >
> > > > > > > 3. The first two experiments in our paper explore the ability of GenAgg vs the baselines to learn useful aggregations.

---

### Official Review · Reviewer_3j4u · 2023-07-08

**Soundness:** 3 good
**Presentation:** 2 fair
**Contribution:** 2 fair
**Rating:** 5
**Confidence:** 4

**Summary:**

The paper proposes a generalized aggregation operator, GenAgg, for GNNs that can significantly improve performance in various tasks. The authors argue that existing aggregation methods are limited in their expressiveness and propose GenAgg as a more flexible alternative. They demonstrate the effectiveness of GenAgg on some benchmark datasets. Overall, the paper makes a contribution to the field of GNNs by introducing a new aggregation operator.

**Strengths:**

- The paper proposes a new aggregation operator, GenAgg, that provides a more flexible alternative to existing methods.
- The authors provide a thorough evaluation of GenAgg on several benchmark datasets and show that it outperforms existing methods.

**Weaknesses:**

- The technical content of the paper is quite limited (only taking 2 pages)
- The paper does not compare with DeepSet like aggregator, both in terms of theory and experiments, which is proved to be fully injective.
- The paper could benefit from an ablation study of GenAgg by showing the results of different parameterizations of f.
- The figures and tables are not professional. They have fonts that are too small to read. Figure 2 should be a table, not a figure.
- The paper could benefit from a more detailed discussion of the limitations of the proposed approach, for example, the computational overhead in terms of memory and runtime.

**Questions:**

- How does the computational complexity of GenAgg compare to existing aggregation methods?
- Is the learned parameters in GenAgg interpretable? Any analysis can be made?

**Limitations:**

- The paper could benefit from a more detailed discussion of the limitations of the proposed approach, for example, the computational overhead in terms of memory and runtime.

---

> ### Author Rebuttal · Authors · 2023-08-07
>
> Thank you for your review, you bring up some good points. We will first address the weaknesses:
>
> - Yes, the method section is about 2 pages—the architecture itself is relatively simple (we regard this as a good thing). However, the rest of the paper also contains technical content, such as the experimental approach and the appendix. For example, we regard Theorem 3.1 (referencing Appendix A) as a particularly interesting and useful theoretical contribution.
> - We do not compare against DeepSets because it is widely regarded as a set function, not an aggregation function—it is different from GenAgg and all of the standard aggregators in that it is not independent over the feature dimension (line 77). It is important to define this boundary between set functions and aggregation functions, as otherwise all GNN architectures would be subsumed under aggregators. In this paper, we stress the importance of constraints specifically in aggregation functions. If we were to augment DeepSets with these constraints, the architecture would approach GenAgg, so one could think of GenAgg as the aggregator variant of DeepSets.
> - We do show the performance of a subset of special cases of GenAgg in each experiment (max, sum, and mean, as shown in Figures 2a, 2b, and 3).
> - Thank you, we will fix this. Are there any specific notes that you have about the figures other than their text size?
> - We will include a figure and discussion about the computational and memory overhead in the discussion section.
>
> To address your questions:
> - If you’re referring to theoretical computational complexity, then sum, PowerAgg/SoftmaxAgg, and GenAgg are in the same category. They scale linearly with the number of features or the number of edges. On the other hand, PNA employs a network which operates over a concatenation of multiple flattened outputs, so the number of parameters (and therefore the number of multiplications) grows quadratically with the number of features. This is discussed in the Parameters section of the Discussion.
> - Yes, the learned parameters can provide intuition about the underlying aggregation function (see the Explainability section of the Discussion for a more detailed explanation).
>
> We will edit the figures’ readability, and add a figure and discussion to the limitations section about the computational overhead of each aggregator. In addition to these items, is there anything else you would like us to add to our rebuttal results / revisions in order to increase your score?

---

> > ### Comment · Reviewer_3j4u · 2023-08-21
> >
> > Since you mentioned
> > > it is different from GenAgg and all of the standard aggregators in that it is not independent over the feature dimension (line 77)
> >
> > Could you clarify why using two separate MLPs for f and f^-1 is independent from the feature dimension, while models like DeepSet which transform each element, aggregate, then transform the output is not independent of the feature dimension? I could always define an element-wise transformation in DeepSet transformations, say I use apply MLP over each dimension (essentially an 1-D conv), which is also independent from the feature dimension.
> >
> > Overall, my concern for the paper is that it really resembles DeepSet where a pair of non-linear transformations are applied before and after elementwise summation. Additionally, it gives the method unfair advantages over the baselines, since those baselines (mean, sum, max, ...) do not involve trainable parameters.
> >
> >
> > In the paper, the claim was
> > > Second, while the learnable functions  and f in Deep Sets are fully connected over
> > the feature dimension, the f and f 211 1 modules in our architecture are scalar-valued functions which are applied element-wise
> >
> > But if you check the DeepSet paper, they describe the transformation as general non-linear functions, and does not restrict them to be fully connected. So your implementation could be regarded as a special implementation of DeepSet.
> >
> > Given these reasons, I would like to keep my evaluation as a weak reject.

---

> > > ### Author Response · Authors · 2023-08-21
> > > **Reply to Reviewer 3j4u**
> > >
> > > There are three main differences between GenAgg and DeepSets:
> > > • The inner and outer nonlinear transformations are inverses of each other (this does not affect whether or not it is independent of the feature dimension)
> > > • It uses per-feature transformations. Using such a transformation in DeepSets would make it more similar to GenAgg.
> > > • It introduces the alpha and beta parameters, which control the dependence on the cardinality and the moment of the distribution.
> > >
> > > We acknowledge in the paper that GenAgg resembles DeepSets, giving it representational power. However, the differences listed above provide useful strong constraints.
> > >
> > > The PowerAgg, SoftmaxAgg, and PNA baselines have trainable parameters as well. With a size 64 feature dimension, PNA adds over 49,000 trainable parameters per layer, while GenAgg adds less than 50.
> > >
> > > Yes, GenAgg is a constrained special case of DeepSets. However, the goal of GenAgg is to provide a constrained aggregation function that can represent any of the standard aggregators (e.g. sum, mean, max, etc...).

---

> > > > ### Comment · Reviewer_3j4u · 2023-08-21
> > > >
> > > > Thanks for the additional explanations. That makes things clearer to me. I'd like to raise my score to 5.
> > > > > With a size 64 feature dimension, PNA adds over 49,000 trainable parameters per layer, while GenAgg adds less than 50.
> > > > Could you explain how this is computed? I thought this analysis was very useful, but it was not discussed in the paper.

---

> > > > > ### Author Response · Authors · 2023-08-21
> > > > > **Reply to Reviewer 3j4u**
> > > > >
> > > > > Thank you!
> > > > >
> > > > > Sure. You're right, in the paper we only briefly allude to the scaling of the parameters in deep learning methods, so we can provide some clarification. PNA calculates the aggregation for 4 aggregators (mean, min, max, std) with 3 different scaling factors (attenuation, identity, amplification), for a total of 12 aggregators. This is concatenated, and there is a linear layer mapping a dimension of size 12d to d, for a total of 12d^2 + d parameters (linear weights plus bias). For a feature size of d=64, we get 12*(64)^2+(64) = 49,216. In practice, we use this feature size of 64 in all of our hidden GNN layers.

---

> > > > > > ### Comment · Reviewer_3j4u · 2023-08-21
> > > > > >
> > > > > > I see. Then how does GenAgg add less than 50?

---

> > > > > > > ### Author Response · Authors · 2023-08-21
> > > > > > > **Reply to Reviewer 3j4u**
> > > > > > >
> > > > > > > GenAgg has less than 50 because the number of parameters is independent of the feature dimension size, as f is applied to each feature dimension individually. The exact number of parameters will depend on the layer sizes, but generally this number will be less than 50. For example, an f with layer sizes [1,2,2,4] (and [4,2,2,1 in f^-1) will result in 22 parameters for f and additional 19 in f^-1, plus two for alpha and beta, for a total of 43 (although the inverse constraint between f and f^-1 further limits the number of free variables, so functionally it is half of that). We can add this calculation to the paper.

---

### Official Review · Reviewer_rgaT · 2023-07-27

**Soundness:** 3 good
**Presentation:** 4 excellent
**Contribution:** 3 good
**Rating:** 7
**Confidence:** 3

**Summary:**

EDIT: Due to author responses below, I have increased my score from a 6 to a 7.

This work proposes a generalized method for parameterizing the aggregation operation of a message passing neural network (i.e. a graph neural network, GNN). This relies on a modification of the generalized $f$-mean that adds two hyperparameters, $\alpha$ and $\beta$. The authors describe how their method, GenAgg, can approximate commonly used GNN aggregators by varying the hyperparameters and the definition of the function $f$. To overcome issues with defining analytical forms of  $f^{-1}$, the authors propose to implement $f$ and $f^{-1}$ with two separate MLPs that learn the inversion.

Then they evaluate GenAgg with three sets of experiments. One tests how well their GenAgg method can approximate common aggregation functions. Then they insert GenAgg into a GNN to see if these results hold true when the aggregator is inside a GNN architecture. Finally, they evaluate a GNN with GenAgg on widely-used benchmarks for GNNs.

**Strengths:**

- The described method and parameterized implementation appear sound and practical for researchers optimizing GNN model designs.
    - In particular, the formulation of the f-mean to overcome various constraints and the demonstration of stable training behavior make this method attractive for GNN practitioners.
- The proposed GenAgg method clearly and stably outperforms the other aggregator methods evaluated in experiments presented in the main text.
- The experiments are well-designed and adequately thorough for this venue.
- The analysis of their experimental results is interesting and insightful (e.g. for the Aggregator regressions, the authors discuss where PowerAgg fails and why theory deviates from practice)
- The authors share their code. Upon first inspection the release appears to be comprehensive, and providing a ready implementation is a useful contribution.

I cannot comment on the novelty because I am not familiar with other recently proposed methods for aggregation in GNNs. While I checked the math throughout the main text, I did not review all the parameter settings for the aggregation functions in Table 1.

**Weaknesses:**

- Some claims are made about the sample efficiency of other methods (e.g. lines 47, 194) that should be supported by citations or data.
- While I do appreciate the inclusion of the data in Appendix D, there should be a more in-depth discussion of the method's limitations in the main text.
    - I am not fully convinced of the overfitting claim, especially because the pattern of results on PUBMED does not differ much from those presented in Figure 3. If overfitting is the reason that GenAgg does not show strong performance benefits on the smaller datasets in Appendix D, we should see a decrease in its benefit when reducing the size of a dataset in the main text (e.g. PATTERN).

**Questions:**

- On lines 145-146, it says that "we found mapping to a higher intermediate dimension can sometimes improve performance over a scalar-valued $f$" with a reference to Appendix E. This claim is not mentioned or alluded to at all in Appendix E.
- Line 151: is $c$ a scalar?
- Section 6 claims that PowerAgg and SoftmaxAgg training curves plateau at least once (lines 326 - 328). However, I do not notice a difference between these training curves and the others. Can you show this more clearly?
- The plots in Fig 3 are hard to read because the lines and error bars often overlap a lot.
    - Please add standard deviation to the table as well, to match the error bars in the plots.
    - I also recommend putting a color coded label on the right side of the plot to indicate the ordering of the final test accuracy (e.g. from top to bottom for CLUSTER: GenAgg, Softmax-agg, Powermean-agg, mean, max, ...). While this information is in the table, it would make the plots easier to read.

Minor typos:
- "the a" - line 71
- "are a quantitative measures" (line 110) -- "a" should be dropped


**Limitations:**

The authors discuss some limitations of the method in Section 6. However, as noted above, including some of the discussion from Appendix D in the main text would be more appropriate.

---

> ### Author Rebuttal · Authors · 2023-08-07
>
> Thank you for your thoughtful review! We will address the weaknesses:
>
> - The closest thing to a DeepSets-based GNN in prior literature is the original MPNN. Despite its representational capacity, subsequent research has focused on more constrained methods. A direct comparison can be seen in the PNA paper [1]. Since it can represent anything that the more recent methods can in theory, but not in practice, we say it has “poor sample efficiency”. For LSTM aggregation, we say it has poor sample efficiency because it is not permutation-invariant (which is acknowledged in the paper where it was introduced). As for PNA in line 194, our argument about the sample efficiency can be backed up by our results in the experiments section.
> - The discussion about overfitting in the limitations section is based on 1) the observation that GenAgg tends to exhibit a larger performance boost on larger datasets, and 2) the observation that the datasets where GenAgg did not provide as much of a performance increase tended to have a large gap between train and test accuracy. These are the main differences between datasets like PUBMED and those like PATTERN. You are right, if overfitting is an issue, we should see a performance decrease if we significantly shrink the size of the PATTERN training dataset.
>
> And your questions:
> - This is alluding to the hidden sizes which are discussed in Appendix E ([1,2,2,4] for $f$, and [4,2,2,1] for f$^{-1}$). If we used the standard formulation for GenAgg, the networks would look something like [1,4,1] (mapping from $\mathbb{R} \mapsto \mathbb{R}$).
> - Yes, $c$ is a scalar, and $x_i$ is as well. Since the feature dimensions are independent, we can consider a feature dimension of size 1 without loss of generality.
> - Most of our training curves are in the Appendix. In this claim, we are specifically referencing Figure 5 in Appendix B.
> - Thank you for the suggestion, we will do this.
>
> [1] Gabriele Corso, Luca Cavalleri, Dominique Beaini, Pietro Lio, and Petar Velickovic. Principal neighbourhood aggregation for graph nets. Advances in Neural Information Processing Systems, 33, 2020.

---

> > ### Comment · Reviewer_rgaT · 2023-08-10
> > **Reply to author comments**
> >
> > Thank you for the responses.
> >
> > Here are point-by-point replies to the weaknesses section:
> > * My point with line 47 is that the phrase "they [PNA/LSTMAgg] are extremely complex" is not a very clear justification for the claim of poor sample efficiency. Your elaboration here is a bit better. I would recommend changing the language to something like "while these models are theoretically universal approximators over set functions, in practice they fail to learn tasks from standard datasets (i.e. are not sample-efficient)." My point for line 194 is that you are not referring to actual data in your paper -- it simply says that the high dimensionality "*can* decrease sample efficiency". Suggestion is to rephrase & give a direct reference to the data, e.g. "appears to decrease sample efficiency (Figure X)."
> > * Are you able to run the experiment of shrinking the size of the dataset? It would be useful to include.
> >
> > Thank you for the replies to my other questions. My concerns there have been addressed.

---

> > > ### Author Response · Authors · 2023-08-10
> > > **Reply to Reviewer**
> > >
> > > You are right that the paper could have been clearer on lines 47/194, we will address this in our revisions. Thank you for the suggestions about phrasing as well, we will integrate those.
> > >
> > > We will try to run the additional experiment that you suggested before the review period is over.

---

> > > ### Author Response · Authors · 2023-08-11
> > > **New Experiment Results**
> > >
> > > We have run the additional experiment that you requested. Unfortunately, OpenReview is not allowing updates to the PDF, so we will summarise the experiment in words:
> > >
> > > Results:
> > > We show plots of the train and test accuracy from three different experiments: the original PATTERN dataset, a reduced version with 10% of the data, and a reduced version with 1% of the data.
> > >
> > > |                     | GenAgg | Best Baseline | Median Baseline |
> > > |---------------------|--------|---------------|-----------------|
> > > | Original, Train Acc | 0.915  | 0.872         | 0.841           |
> > > | Original, Test Acc  | 0.926  | 0.897         | 0.866           |
> > > | 10%, Train Acc      | 0.958  | 0.903         | 0.846           |
> > > | 10%, Test Acc       | 0.905  | 0.854         | 0.829           |
> > > | 1%, Train Acc       | 0.978  | 0.846         | 0.832           |
> > > | 1%, Test Acc        | 0.903  | 0.832         | 0.830           |
> > >
> > > Discussion:
> > > As expected, reducing the size of the dataset leads to overfitting. As the size of the dataset decreases, the gap between GenAgg’s train and test accuracy widens. Somewhat unexpectedly, a significant performance gap remains between GenAgg and the baselines even when training on a dataset which is 1% of the original size. This begs the question that you raised in your review: what is the difference between datasets like PATTERN and PUBMED, and how do those differences correlate with the performance of GenAgg? This new experiment highlights one interesting difference: in the smaller datasets of coauthor data, all of the methods overfit (i.e. the train accuracy goes to 1), whereas in the reduced PATTERN dataset, the train accuracies of the baselines do not increase past a certain point. If we were to speculate about these findings, we would say that GenAgg is less likely to provide a performance boost in datasets where a simple GNN architecture already provides enough complexity (e.g. the coauthor datasets). On the other hand, given that the baselines’ train accuracy plateaus on the reduced PATTERN dataset while GenAgg’s train accuracy goes to 1, it is likely that there is something fundamental about the dataset that requires a more complex aggregation function to represent.

---

> > > > ### Comment · Reviewer_rgaT · 2023-08-16
> > > > **Reply to new experiment results**
> > > >
> > > > Thank you for this follow-up. The results are indeed quite interesting, and I suggest that the new interpretation ("there is something fundamental about the dataset that requires a more complex aggregation function to represent") should be added to the main discussion of the paper. The data for this additional experiment can be added to the Appendix, since there is already enough content in the main paper.
> > > >
> > > > The reason I advocate to add this interpretation is that I think it is a useful contribution to the literature on aggregation functions in GNNs. It also matches my previous experience training and testing on GNN benchmarks as well -- there is something about some datasets/tasks/graphs which requires a more complex model, but right now there is no formal explanation for why this is the case.
> > > >
> > > > I appreciate the authors' responsiveness to my concerns. As a result I will increase my score to a 7.

---

> > > ### Author Response · Authors · 2023-08-13
> > > **Reply to Reviewer**
> > >
> > > If all of your questions/concerns have been addressed, would you consider increasing your score?

---

### Author Rebuttal · Authors · 2023-08-07

Two reviewers have requested that we add a table about the computational overhead of GenAgg to our Limitations section. We include our results in the attached PDF.

---

### Decision · Program_Chairs · 2023-09-21

**Decision:**

Accept (poster)

**Comment:**

After the additional experiments performed during the rebuttal period all reviewers voted for acceptance. The authors should include the additional experiments and apply the proposed changes in the revised version of the manuscript.